# Mapping human pre-rRNA processing and modification at single nucleotide resolution using long read nanopore sequencing

Stefan Pastore[1], Ludivine Wacheul [2], Lioba Lehmann[3], Stefan Mündnich [1], Beat Lutz[4,5], Mark Helm [1], Susanne Gerber [3,6], Denis L. J. Lafontaine [2] & Tamer Butto [1]

Ribosome biogenesis requires the synthesis and sequential processing of precursor rRNAs (pre-rRNAs) into mature rRNAs. Traditional methods such as northern blotting and metabolic labeling provide limited resolution. Here, we present NanoRibolyzer, a nanopore-based long-read sequencing approach that enables ab initio identification and quantification of rRNA precursors while simultaneously mapping RNA modifications. Using supervised and unsupervised mapping, we detect both known and previously uncharacterized pre-rRNAs and delineate cleavage events at single-nucleotide resolution. A simple cell-fractionation protocol further separates nuclear and cytoplasmic pre-rRNAs, allowing spatial deconvolution of processing pathways. By projecting each sequenced molecule in a two-dimensional space using its starting and ending coordinates, we generate an intuitive representation in which the activity of the 5′ → 3′ and 3′ → 5′ exoRNases can be tracked as they mature pre-rRNAs one nucleotide at a time. Targeted knockdowns of ribosome-assembly factors quantify accumulation of intermediates and reveal condition-specific processing "fingerprints" with biomarker potential. High-resolution re-analysis of known factors uncovers unexpected functions. Additionally, pseudouridine mapping shows that the primary 47S transcript is extensively modified, whereas aberrant intermediates (34S and 36S-C) are hypomodified. With its high resolution and unique discovery mode, NanoRibolyzer provides new insights into rRNA processing and modification, greatly advancing our understanding of ribosome biogenesis.

Ribosomes are ribonucleoprotein nanomachines responsible for protein synthesis in all living cells[1,2]. Ribosome biogenesis is a complex process involving the synthesis, processing, and modification of precursor ribosomal RNAs (pre-rRNAs), as well as RNA folding and packaging into functional ribosomal subunits. In eukaryotes, this pathway is initiated in the nucleolus, where a large ribosomal RNA precursor (pre-rRNA), the 47S, is synthesized by RNA polymerase I (Pol I)[3,4]. The 47S contains sequences for three out of four rRNAs (the 18S,

[1]Institute of Pharmaceutical and Biomedical Sciences, Johannes Gutenberg-University Mainz, Mainz, Germany. [2]RNA Molecular Biology, Fonds de la Recherche Scientifique (F.R.S./FNRS), Université libre de Bruxelles (ULB), Biopark Campus, B-6041 Gosselies, Belgium. [3]Institute of Human Genetics, University Medical Center of the Johannes Gutenberg University Mainz, Mainz, Germany. [4]Leibniz Institute for Resilience Research (LIR), Mainz, Germany. [5]Institute of Physiological Chemistry, University Medical Center Mainz, Mainz, Germany. [6]Institute for Quantitative and Computer Biosciences (IQCB), Mainz, Germany. e-mail: denis.lafontaine@ulb.be; buttamer@uni-mainz.de

5.8S, and 28S) interspersed with non-coding spacers (Fig. 1A and S1). The fourth rRNA, 5S, is produced independently by Pol III, in the nucleoplasm. Following transcription, pre-rRNAs undergo a series of maturation steps, including processing (cleavage), modification, and packaging with ribosomal proteins, to release the mature rRNAs and produce the ribosomal subunits, which are ultimately exported to the cytoplasm where they engage in translation[5]. Throughout this multi-step process, the nascent transcripts undergo extensive processing by endonucleases performing precise cleavages within the external and internal transcribed spacers (ETS and ITS, respectively), often followed by exonucleases that progressively trim pre-rRNAs, ultimately releasing the mature rRNAs (Fig. 1A and S1). The progressive trimming of pre-rRNAs by exonucleases results in the production of transient, metastable species that likely remain largely uncharacterized. Additionally, these processes contribute to the generation of poorly defined RNA ends, further highlighting the complexity and incomplete understanding of pre-rRNA processing. Disruptions occurring at any stage of

the pathway may activate regulatory cascades, including surveillance leading to the accumulation of distinctive intermediates, which can significantly impact ribosome function, cellular protein synthesis, and overall cellular homeostasis[4,7].

More than two decades of research have led to major advances in identifying discrete processing sites and pre-rRNA intermediates, which now serve as critical markers for assessing ribosome biogenesis efficiency (Fig. 1A and S1). Knowledge of these intermediates has been particularly valuable for investigating aberrant precursor production that arises during processing perturbations[2]. Conventional approaches for analyzing rRNA processing intermediates, such as northern blotting, metabolic labeling, or primer extension, allow for the identification of these accumulated precursors and cleavage sites, respectively[6,8–10]. However, these assays require important input material (often µg range) and have limited resolution and throughput. Additionally, studying pre-rRNA processing by sequencing has remained challenging due to the highly repetitive nature of rDNA

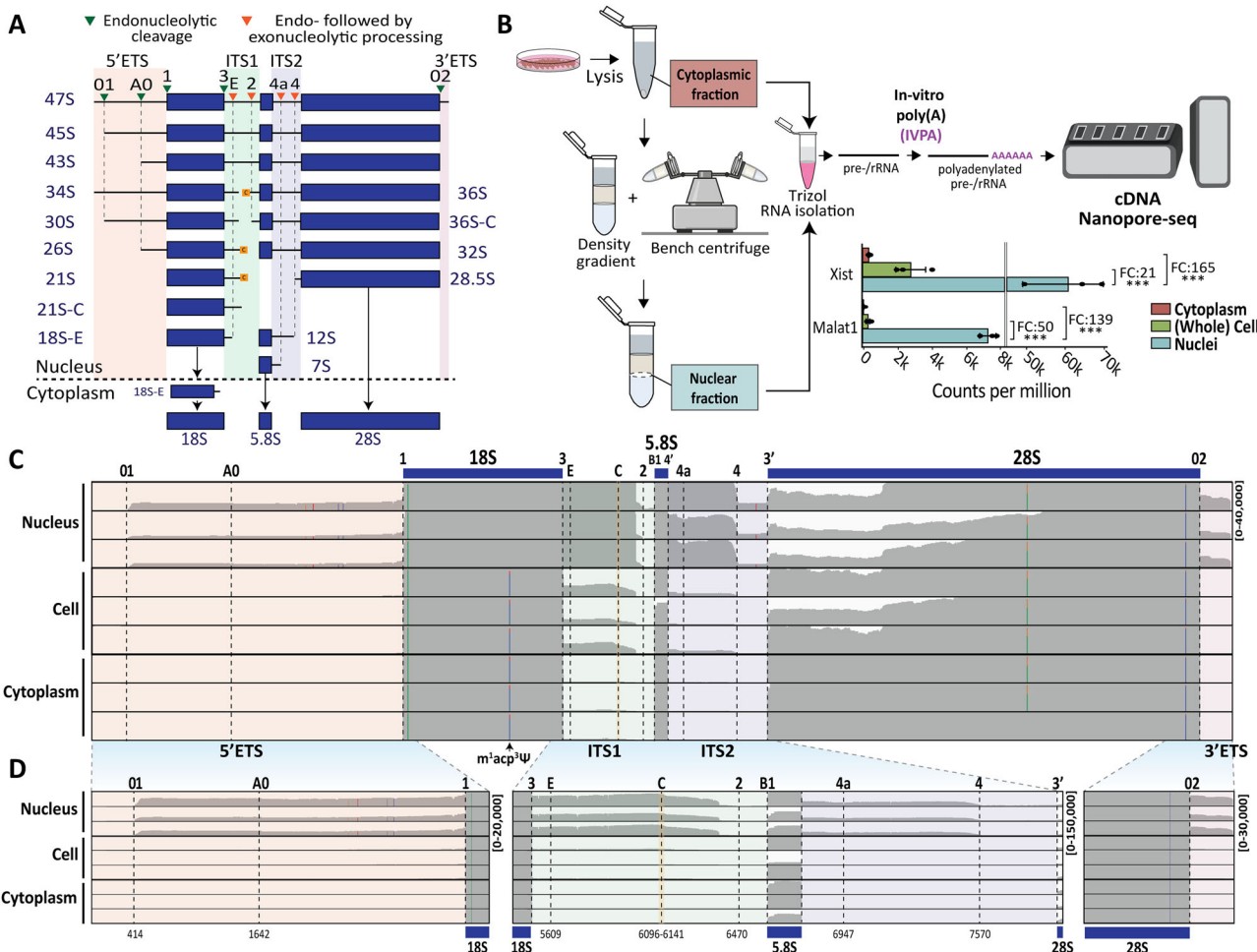

**Fig. 1 | Streamlined nuclei isolation procedure. A** Simplified pre-rRNA processing pathway in human cells. Three of the four mature rRNAs (18S, 5.8S and 28S), produced as a single long polycistronic precursor, 47S, by RNA polymerase I, are interspersed with the 5′ and 3′ external (ETS) transcribed spacers and internal transcribed spacers (ITS) 1 and 2. The cleavage sites (01, A0, 1, etc.) and main processing intermediates are indicated. Some species (e.g., 34S, 36S, 36S-C) are associated with aberrant processing. **B** Nuclei isolation procedure for isolation of nuclear and cytoplasmic RNA followed by cDNA nanopore-seq (see "Materials and methods" and Supplementary Fig. 1 for details). Normalized read count per million of *Xist and Malat1* transcripts in cytoplasmic (cyt), whole cell (cell), and nuclear (nuc) fractions (n = 3 samples each). Fold change (FC) between the conditions is shown above each comparison. One-way ANOVA followed by Tukey test for

multiple comparison post hoc test, *Xist*: P(nuc-cyt) = 4.94e-05; P(nuc-cell)=6.29e-05; P(cell-cyt)=0.878. *Malat1*: P(nuc-cyt)=3e-07; P(nuc-cell) = 4e-07; P(cell-cyt) = 0.536. ***p < 0.001. **C** IGV coverage profiles of representative nuclear, whole cell and cytoplasmic samples across 47S. The data range was normalized to 40,000 across all samples to visualize the coverage profiles within the selected regions. The m¹acp³Ψ modification in 18S, which is visible in the cytoplasm but not in the nucleus, is highlighted. Positions with a mismatch frequency higher than 0.2 are colored. **D** Zoom in IGV coverage profiles across 5′ ETS and 18S (left), ITS1, 5.8S and ITS2 (middle) and 28S and 3′ ETS (right) of representative nuclear, whole cell and cytoplasmic samples across 45SN1. The data range shown on the right of each figure was normalized across all samples to visualize the coverage profiles within the selected regions.

arrays and poor genome annotations, making it difficult to accurately map reads and distinguish between individual rDNA copies, especially with short-read sequencing technologies[11–14].

Nanopore sequencing (nanopore-seq) has emerged as a promising technology to investigate ribosome biogenesis[15–17]. The key advantage of nanopore-seq lies in its ability to sequence long reads, such as cDNAs, as well as native RNA molecules via direct RNA sequencing (DRS), allowing for the investigation of entire transcripts, including their modifications[18,19]. These capabilities are particularly valuable for studying rRNA intermediates, which can vary significantly in length and abundance, and possibly modification levels. To this date, there are no tools that exploit long-read sequencing for the analysis of human ribosomal RNA precursors.

Here, we present NanoRibolyzer, a method that integrates state-of-the-art long-read nanopore sequencing with advanced bioinformatics to achieve spatially resolved, single-nucleotide analysis of pre-rRNA intermediates. Using a streamlined nuclei isolation protocol, we systematically profile precursor and mature rRNA species in both nuclear and cytoplasmic compartments. We further introduce precursor-specific modification analysis, uncovering the spatiotemporal dynamics of rRNA modifications. Overall, NanoRibolyzer enables comprehensive detection and quantification of both known and novel processing intermediates, including cleavage events generated by endo- and exoribonucleolytic activity, as well as pseudouridine modifications.

## Results

### Nuclear isolation enabling high-resolution mapping of pre-rRNA precursors

Ribosome biogenesis begins in the nucleolus, a multiphase condensate within the nucleus, continues in the nucleoplasm, and is finalized in the cytoplasm, where mature rRNAs accumulate. In order to develop a nanopore-sequencing approach to study pre-rRNA processing, we adapted a straightforward isolation protocol involving density gradient separation using a simple benchtop centrifuge (Fig. 1B and S2). This provided spatial resolution of processing events and access to purified nuclear RNA enriched in low-abundance, short-lived pre-rRNAs that are otherwise masked in total RNA by the highly abundant mature cytoplasmic rRNAs. We applied the protocol to HEK293 cells to produce highly purified nuclear and cytoplasmic RNA fractions, and as a control, whole cell total RNA ("whole cell") (Fig. 1B and S2A).

After separation of the fractions, a quality assessment of isolated nuclei was conducted using DNA staining with DAPI, revealing debris-free and round intact nuclei[20] (Fig. S2C). RNA was extracted and electropherograms produced using a TapeStation. As expected, the whole cell and cytoplasmic fractions displayed the abundant mature 18S and 28S rRNAs (all analyses performed in triplicate throughout this work, R1-R3) (Fig. S2D). In contrast, the nuclear fractions exhibited higher molecular weight species, at the expected size for pre-rRNAs (Fig. S2D). Northern blot analyses performed throughout the study consistently verified that precursor rRNA is predominantly retained within nuclear fractions, in contrast to cytoplasmic and whole-cell fractions (Fig. S3, Source Data 1).

Since Nanopore-based RNA library preparation strategies typically capture poly(A) + RNA[21], we applied an in vitro polyadenylation strategy followed by long-read cDNA sequencing (Fig. 1B). To assess enrichment of transcripts in the nuclear fractions, we quantified two nuclear long non-coding RNAs (lncRNAs): XIST and MALAT1 (Fig. 1B and Supplementary Table S1). As expected, comparative analysis of the abundance of these transcripts across different cellular compartments revealed several fold-change enrichments in the nuclear fraction relative to the cytoplasm (up to 165-fold for XIST).

Compared with whole-genome alignments, which are challenging to interpret for rRNAs due to the presence of hundreds of rDNA gene copies, NanoRibolyzer aligns long rRNA reads to a single 47S reference template (45SN1; GeneID: 106631777). This approach provides a clear and unambiguous view of reads corresponding to pre-rRNAs and mature rRNAs. We generated an average of ~3 million reads per sample with an average ~75% alignment rate to the 47S template. Our bioinformatic pipeline is depicted in Fig. S4A. For reference and to initiate precursor quantification analysis, we retrieved the positions of known processing sites and major precursors from literature[8–10] (Fig. 1A and S1).

We applied NanoRibolyzer to unperturbed HEK293 cells and characterized reads mapping to mature rRNA sequences and the non-coding spacers (5' ETS, ITS1, ITS2 and 3' ETS) for the nuclear, cytoplasmic, and whole cell fractions (Fig. 1C, D). As expected, the coverage profiles revealed that the nuclear fraction had higher coverage across the spacers (ETSs and ITSs) compared to the cytoplasmic ones, with the whole cell fraction displaying an intermediate coverage level (Fig. 1C, D). The start and ends of these reads corresponded to known processing sites, e.g., sites 01, 1, 2, 4, etc. (see refs. 8–10).

These observations agree with the notion that the nuclear fraction predominantly contains ribosomal assembly intermediates, in line with the fact that most steps of ribosome biogenesis occur in the nucleolus. In our initial analyses, we consistently observed that whole-cell fractions are largely redundant with cytoplasmic fractions (see Fig. 1D, zoomed IGV). Therefore, throughout the remainder of the manuscript, and unless stated otherwise, we focus on comparisons between nuclear and cytoplasmic fractions.

We also noted that the hyper-modified nucleotide 1-methyl-3-α-amino-α-carboxyl-propyl pseudouridine (m$^1$acp$^3\Psi$) at position 1248 on the 18S rRNA was detected in whole cell and cytoplasmic fractions, but strikingly not in nuclear ones (Fig. 1C, arrow), consistent with the fact that m$^1$acp$^3\Psi$ formation is completed in the cytoplasm[22].

### Mapping rRNA precursor diversity and processing pathways

Once we confirmed that nuclear sequencing reads extend across spacer regions, we employed two complementary approaches to quantify pre-rRNAs and resolve processing sites: (1) a rather classical supervised or "template-based" strategy and (2) an unsupervised or "template-free" method (Fig. S4).

The supervised approach detects pre-rRNA intermediates and spacer fragments described in the literature, quantifying their relative abundance by maximizing reciprocal overlap (Fig. 2A and S4B). These intermediates and fragments are defined based on published processing sites (Fig. 1A and refs. 1,2). Quantified data can be visualized either as heatmaps of averaged log10 reads per million (Fig. 2B and S5A) or histograms illustrating averaged reads per million for each condition with associated standard deviations and p-values (Fig. S5B).

Using the supervised approach, as expected, we found that most pre-rRNA intermediates are substantially more abundant in the nucleus than in the cytoplasm (Fig. 2B and Fig. S5A, B; statistical analysis in Source Data 1, *p*-values on histograms). This analysis also highlights that several intermediates historically associated with ribosome biogenesis defects[6], such as the 34S, 36S, or 36S-C, are not, or only marginally, produced in unperturbed cells (see dark shade of blue in the heatmap of Fig. 2B and S5A). As we will show later, when mapping rRNA modifications at the level of individual pre-rRNA precursors, these aberrant species are, strikingly, not faithfully modified. The analysis also reveals that precursors associated with early steps of large ribosomal subunit biogenesis (32S) are not detected in the cytoplasm, while precursors associated with late maturation of the small subunit (21S, 21S-C, 18S-E) and the large subunit (12S, 7S) are partially present in the cytoplasm (Fig. 2B and Fig. S5A, B).

Together, these results demonstrate that NanoRibolyzer provides a quantitative framework to assess the relative abundance of all major pre-rRNA precursors as well as mature rRNAs.

Having assigned long-read sequences to specific pre-rRNA precursors and mature rRNAs, we next sought to pinpoint their

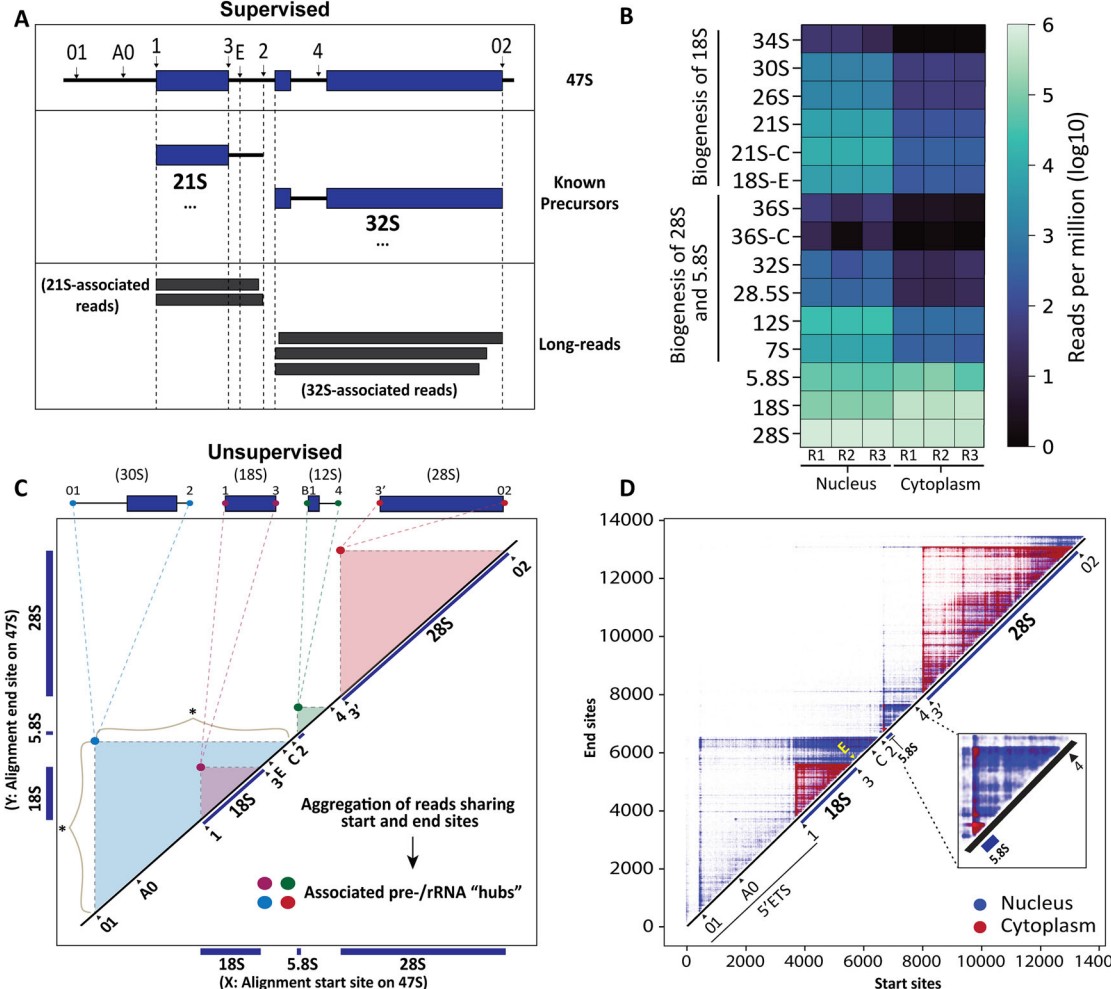

**Fig. 2 | Quantification of rRNA precursors and processing sites using NanoRibolyzer. A** Simplified overview of supervised (template-based) approach using minimal reciprocal overlap (MRO). Query reads are compared to literature-based intermediates (Fig. 1A), and each read is assigned to the intermediate with the highest overlap based on alignment start and end positions. In the example shown, the query reads are closely associated with 21S and 32S precursors. After processing all the reads, the data is presented as a relative quantification score in reads per million, allowing for clear visualization and comparison. See more details in Supplementary Figs. S3 and S4. **B** Quantification of detected pre-rRNA intermediates and mature rRNAs in the nucleus and cytoplasm ($n$ = 3 samples each). **C** Rationale of the unsupervised (template-free) approach. A 2-D matrix representing the

RNA45SN1 template is constructed, with each rRNA read plotted by its start (x-axis) and end (y-axis) positions. This approach maps transcript boundaries and highlights intensity "hubs", which indicate abundant rRNA products near mature rRNAs or putative processing sites. In the example shown, the intensity hubs for 30S, 18S, 12S and 28S correspond to reads clustered at the start and end of the respective pre-/rRNAs. The (*) symbol represents the exoRNase trajectories (see text for details, Fig. S3C). **D** Overlayed intensity matrices of nucleus (blue) and cytoplasm (red), highlighting contrasting read distributions: ETS and ITS-associated reads dominate in the nucleus (in blue), while the cytoplasm predominantly contains mature rRNA reads (in red).

predominant start and end positions, thereby defining processing sites. To do so, we computed significant 5′ and 3′ boundaries from the template-based BAM files, using the 45SN1 (hg38) reference sequence. Sites occurring at least two standard deviations above the mean were classified as significant, and results were exported in TSV and BED formats for visualization (Fig. S5C). To ensure robustness, we retained only those sites present in at least two of the three biological replicates, enabling us to identify the most reproducible processing sites and quantify their prevalence across the 47S sequence (Source Data 1).

Inspection of the cleavage sites revealed consistent positions within the spacer regions and at the boundaries of the mature rRNAs (Fig. S5D; start sites in green, end sites in orange). Cleavages detected in the spacer regions, particularly those proximal to previously reported processing sites, were considered bona fide processing events (Fig. S5D–F; see also Fig. 1A and Fig. S1). In several cases, using the literature information alone was insufficient to recognize with single-nucleotide precision the RNA end points. As discussed below,

this limitation will be solved by using our complementary unsupervised mapping approach (Fig. 2C, D and Fig. 3).

As expected, samples from the cytoplasm, where pre-ribosomes are nearly fully matured and where mature ribosomes are engaged in translation or storage, showed cleavage sites mostly restricted to the extremities of mature rRNAs (Fig. S5D). In sharp contrast, nuclear samples displayed numerous internal cleavages within the spacer regions, reflecting ongoing pre-rRNA processing (Fig. S5E, F).

We also found numerous cleavages within mature rRNA sequences (Fig. S5D). These aberrant events, which generate by-products destined for degradation, have been greatly underappreciated using classical approaches such as northern blotting. Their relevance becomes clear below, where we show they can define condition-specific "fingerprints" (e.g., upon URB1 or RPL3 depletion, see Fig. 4E), with potential biomarker value.

In conclusion, the supervised component of NanoRibolyzer performs very well to quantify the abundance of RNA species previously

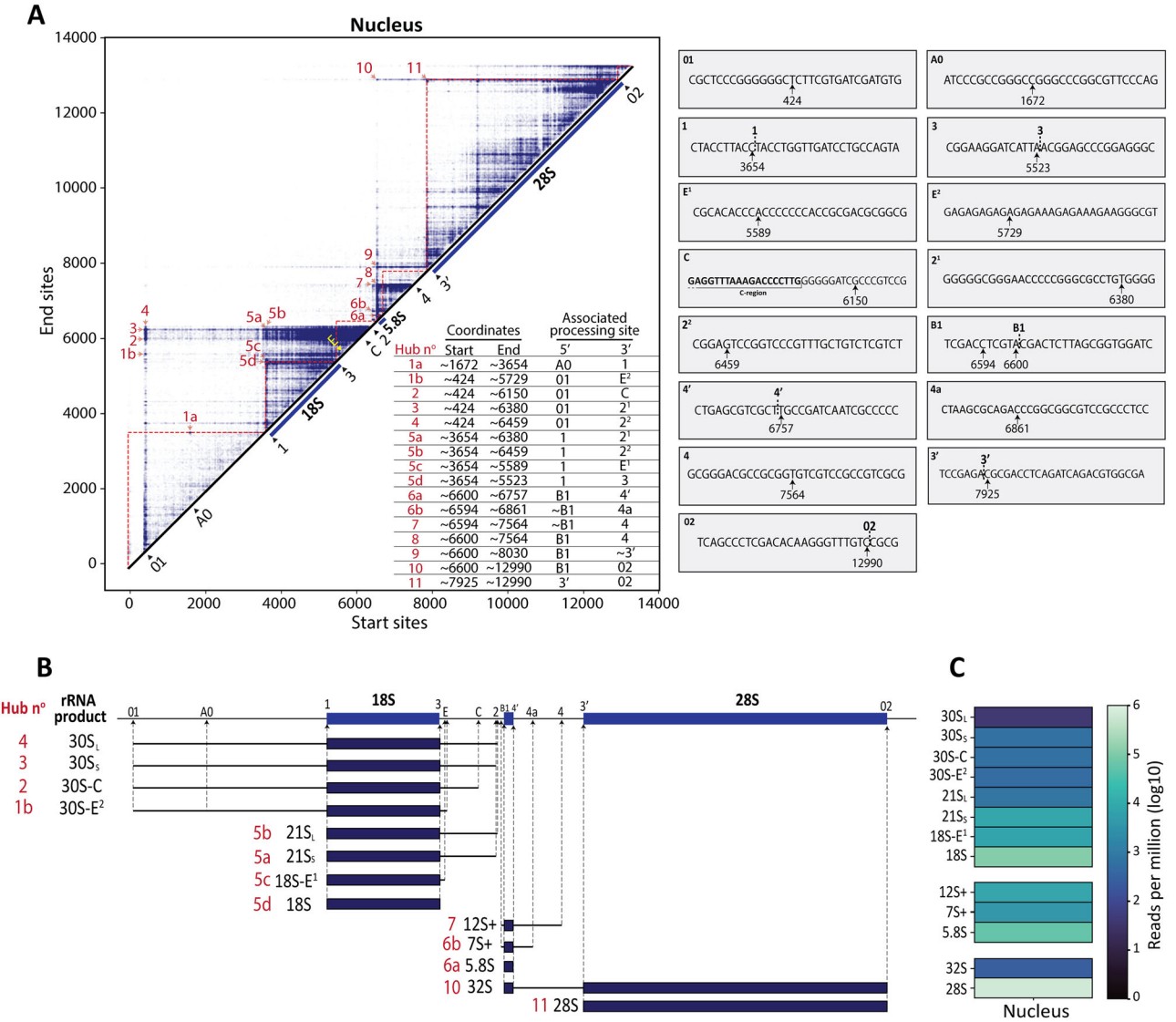

**Fig. 3 | Identification of novel processing sites and precursors with nucleotide resolution. A** Left, Intensity matrix of nuclear reads, with intensity "hubs" numbered in red. The "hubs" were identified based on intensity thresholding and closeness to bona fide processing sites. Right, Details of each mapped processing site. **B** Illustration of the identified rRNA products in the nucleus, as derived from the intensity hubs table shown in panel A. The corresponding rRNA products are shown on the left, together with the hub number (in red) and associated precursor (black). In hubs 6b and 7, the (+) symbol depicts extensions 5'upstream 5.8S. **C** Quantification of newly detected nuclear pre-rRNAs (details in Fig. S9B).

described in the literature and, with few exceptions, to map their ends with single-nucleotide resolution. In essence, it performs quite similarly to northern-blotting, except that it has additionally started to reveal an under-appreciated level of cleavage within mature sequences.

Nonetheless, the supervised approach is not suitable to map the low-abundant metastable (transient) species, which results from the progressive exoribonucleolytic cleavage, 5'→3' and 3'→5', which are known to be of paramount importance in processing, generating thousands of intermediates. To put it simply, a huge diversity of species still escapes characterization.

To address this limitation, we next implemented an unsupervised approach that provides a completely unbiased mapping strategy, i.e., a method that does not rely on any preconceived knowledge about pre-rRNA processing. We also developed a new display to apprehend processing intuitively.

In this approach, every single individual read sequenced is projected onto a two-dimensional matrix (Fig. 2C, D and Fig. S4C). The *x*-

and *y*-axes extend from the transcription start site to the termination site of the primary 47S transcript, respectively. Each rRNA read is positioned on the matrix according to its 5' and 3' coordinates, enabling precise, reference-free mapping of transcript boundaries (Fig. 2C, D and Fig. S4C). This visualization reveals intensity "hubs" corresponding to abundant rRNA species whose ends coincide with established or putative novel processing sites (Fig. 2C, D and Fig. S4C; see "Methods" for details).

We applied the unsupervised approach to nuclear and cytoplasmic fractions of unperturbed HEK293 cells to visualize the unbiased aggregation of reads associated with the 47S transcript. In the nuclear fraction, we observed prominent accumulations of reads within the ETS and ITS regions, corresponding to processing sites and appearing as characteristic blue "pyramids" in the matrix (Fig. 2D). For example, striking pyramids were detected between site 01 in the 5' ETS and site 2 in ITS1, defining precursors of 18S rRNA (see below), and between the 5' end of 5.8S and site 4 in ITS2, corresponding to 3'-extended 5.8S precursors

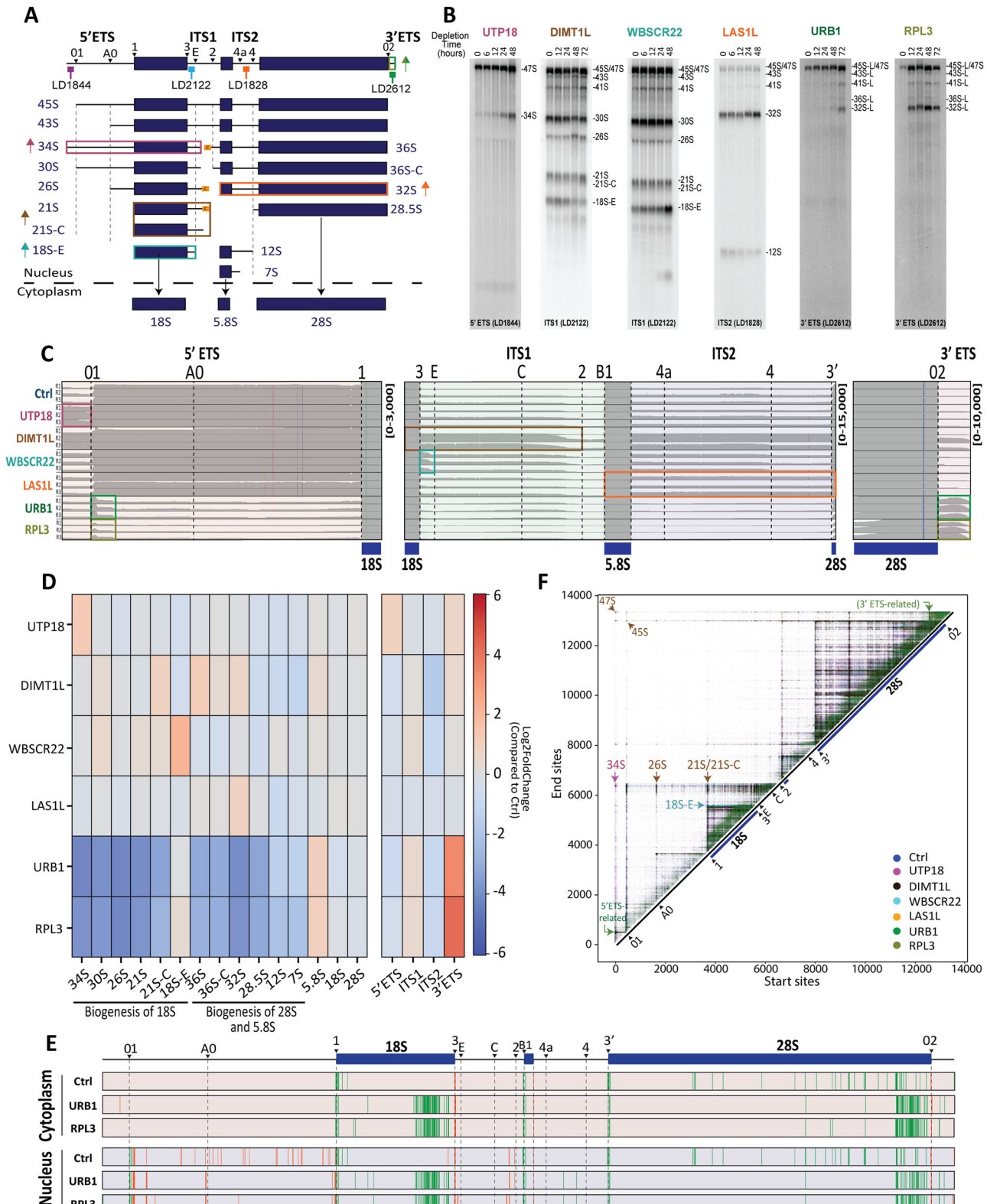

destined for $3' \rightarrow 5'$ trimming by the multi-subunit RNA exosome[6,23] (zoomed inset in Fig. 2D).

Remarkably, this approach also visualizes, the activity of $5' \rightarrow 3'$ and $3' \rightarrow 5'$ exoRNases as they progressively trim spacer sequences one nucleotide at a time (Fig. S6A, B). Low-abundance, metastable intermediates generated during this process, which are difficult to detect with conventional methods, align as continuous signal traces in the matrix: horizontal lines reflecting $5' \rightarrow 3'$ exonucleolytic trimming, and

vertical lines corresponding to $3' \rightarrow 5'$ exonucleolytic activity (Fig. S6A, B). Remarkably, the positions where the exoRNases pause can be seen as more intense dots.

In contrast, reads from the cytoplasmic fraction (shown as "red pyramids," in Fig. 2D, Fig. S6C) mapped almost exclusively to the three mature rRNAs, 18S, 5.8S and 28S (Fig. 2; see inset for 5.8S). The top of each pyramid corresponds to the actual mature RNAs, whereas the interior signals to abundant degradation products. Again, such an

**Fig. 4 | Analysis of pre-rRNA processing following perturbations using NanoRibolyzer. A** Simplified schematics of processing with the position of probes used in Northern blotting in the primary transcript (LD1844, LD2122, LD1828, and LD2612). RNA intermediates most affected upon factor knockdown highlighted with the following color-code: UTP18, purple; DIMT1L, brown; WBSCR22, cyan; LAS1L, orange; URB1, dark green; RPL3, khaki. **B** Defining a calibration set of reference factors to benchmark Nanoribolyzer. Factor depletion was assessed by Northern blotting. Each factor was depleted in HEK293 cells in a time course (from 6- to 72-h). The optimal depletion time point was selected for NanoRibolyzer analysis. For UTP18, WBSCR22, and LAS1L it was 48 h. For DIMT1L, URB1, and RPL3, it was 72 h. **C** IGV coverage profiles of nuclear reads of control and knockdown of UTP18, DIMT1L, WBSCR22, LAS1L, URB1 and RPL3 across the 5′ ETS (left), ITS1, 5.8S, and ITS2 regions (middle) and 3′ ETS (left). The transcript regions affected reflect the processing perturbations and are color-coded like in panel A. **D** Quantification of Log2FoldChange in nuclear pre-rRNA intermediates and mature rRNAs upon knockdown of UTP18, DIMT1L, WBSCR22, LAS1L, URB1 and RPL3 knockdown samples compared to control ($n = 3$ samples each). **E** Visual representation of the predominant start (green) and end (orange) sites of reads aligned to the 47S rRNA in cytoplasmic (top) and nuclear (bottom) fractions, comparing control samples with URB1 and RPL3 knockdowns. Cleavage sites are shown only if detected in at least two of the three biological replicates. **F** Overlaid intensity matrices of nuclear reads from control, UTP18, DIMT1L, WBSCR22, LAS1L, URB1, and RPL3 knockdown samples. Representative "intensity hubs" are highlighted with arrows and annotated with the corresponding RNA species where applicable.

abundance of mature rRNA fragments was truly unexpected and had not been characterized before. Individual matrices for all three biological replicates are shown in Fig. S6C, underscoring the robustness of the analysis.

In conclusion, combining supervised and unsupervised mapping approaches not only exploits existing knowledge of abundant, well-characterized pre-rRNA intermediates, but also enables the discovery of previously undetected, low-abundance and metastable processing intermediates as well as degradation by-products. Our analysis reveals the progression of exoRNases on pre-rRNAs, highlighting stalling positions. Finally, we also uncover an exceptionally rich and largely overlooked population of aberrantly cleaved species. As shown below, these aberrant products carry powerful biological information: depletion of factors acting at the same step of ribosome biogenesis yields highly similar patterns of cleavage by-products (Fig. 4E), effectively generating condition-specific molecular fingerprints.

### Redefining pre-rRNA processing sites to single-nucleotide resolution
Next, we followed an agnostic approach to remap the processing sites by analyzing the intensity matrix obtained by unsupervised mapping. To achieve this, we extracted approximate coordinates from detectable intensity "hubs" in the nuclear fraction and annotated the associated 5′ and 3′ processing sites (Fig. 3A). These coordinates should be considered estimates, as minor RNA species often show variability in their end points. We therefore bioinformatically selected the most predominant end points located near putative processing sites, using a threshold of at least ten reads per hub.

Analysis of the intensity matrix revealed obvious cleavages, which could all be mapped to single-nucleotide resolution; many were associated with known processing sites[8–10,24–32], and there were also new sites. The detailed description of all processing sites mapped, novel and known, is provided in Table 1 and Fig. S7. Briefly, we mapped thirteen known processing sites to single-nucleotide resolution, revealing multiple endpoints at two sites ($E^1$ and $E^2$, and $2^1$ and $2^2$). For sites already known in the literature, there was an excellent agreement between sites identified by the supervised and unsupervised approaches (Fig. S5D–F).

In summary, we have confirmed all known processing sites and identified new sites to single-nucleotide resolution, providing a powerful resource to the community for years to come.

### NanoRibolyzer enables quantification of novel pre-rRNAs
The intensity hub mapping strategy (unsupervised approach) led to the identification of novel processing sites, and we next inspected the products resulting from cleavage at these sites (Fig. 3B).
We draw several conclusions:

(1) the detection of precursors starting at site 01, in the 5′ ETS, and ending at different positions in ITS1 (including at site 3, E, C, $2^1$ and $2^2$) indicates that cleavage in ITS1 can occur at least to some extent prior to cleavage at site A0 (Fig. 3B, Fig. S7; Hubs n° 1b and 2-4).

(2) we confirmed that the 21S corresponds to two isoforms[30]: $21S_S$ (small) and $21S_L$ (large) (Fig. 3B, Fig. S7; Hubs n° 5a-b, and S5B).

(3) we confirmed that maturation of the 3′ end of 5.8S involves progressive exoribonucleolytic trimming of 12S to 7S, and ultimately to 5.8S (Fig. 3B, Fig. S7; Hubs 6b-7 and S6B), a process known to involve substrate handover between RNA exosome subunits[6].

Having identified new cleavage sites and intermediates using the unsupervised model (Fig. 3B), we then incorporated this information into the supervised approach to further refine our processing analysis and to formally quantify the newly detected intermediates through reciprocal overlap maximization (see above and Fig. S4B).

This analysis revealed that the shorter pre-rRNAs are systematically more abundant than the larger pre-rRNAs from which they derive, as generally expected in a precursor-product relationship. This was true in the pathways of production of 18S, 5.8S, and, also of 28S (Fig. 3C and Fig. S8A, B).

To further illustrate how NanoRibolyzer can easily be adapted to detect and quantify new pre-rRNA species, we focused on 3′ end maturation of 18S rRNA (Fig. S8C). The biogenesis of 18S rRNA 3′ end is particularly interesting because it initiates in the nucleoplasm, it ends in the cytoplasm, and it involves formation of a succession of metastable forms: the 18S-E extended respectively by 10, 24, 36, 40, and 78 nucleotides, which are progressively trimmed to 18S-E and finally 18S (ref. 26). Once we incorporated this information in our supervised approach (Fig. S8C), it revealed the respective abundance of each intermediate across the nucleus and cytoplasm as the precursors mature across the nuclear pore complexes (Fig. S8D). Depletion of ribosomal protein RPS26 (Fig. S9A–C), which is known to block the final cytoplasmic steps of 18S rRNA maturation while leaving nuclear processing unaffected[33,34], indeed led to accumulation of the 18S-E + 24, +36 and +40 in the cytoplasmic fraction but not in the nucleus (Fig. S8E, F and Fig. S9).

### NanoRibolyzer captures pre-rRNA processing perturbations
Having demonstrated the ability of NanoRibolyzer to capture pre-rRNA processing in unperturbed cells, we next aimed to evaluate how efficiently our method can detect processing perturbations. To achieve this, we selected key processing factors involved in the maturation of each spacer sequence and depleted them (Fig. 4A and Fig. S3).

For the 5′ ETS, we targeted UTP18, a component of the SSU-processome, whose depletion results in accumulation of the aberrant 34S species (Ref. 35). For ITS1, we selected DIMT1L and WBSCR22, whose loss results respectively in accumulation of the 21S/21S-C and the 18S-E pre-rRNAs[36,37]. For ITS2, we selected the endoRNase LAS1L, whose absence causes accumulation of 32S pre-rRNA[38]. Lastly, for the 3′ ETS, we chose URB1, whose importance for 3′ ETS removal has been

**Table 1 | Redefining the human ribosomal processing sites**

| Spacer segment | Processing Site | Reported position of the processing site | Proposed position of processing site (45SN1) |
|---|---|---|---|
| 5′ ETS | O1 | ~$C^{414}$-$C^{416}$ $G^{420}$-$U^{422}$ | $C^{423}$^$U^{424}$ $G^{429}$^$U^{430}$ $G^{443}$^$A^{444}$ |
| | A0 | ~$G^{1643/1642}$ | $C^{1609}$-$U^{1627}$ $C^{1671}$^$C^{1672}$ |
| 5′ ETS (5′ of 18S) | 1 | $U^{3653}$ | $C^{3654}$^$U^{3655}$ |
| ITS1 (3′ of 18S) | 3 | $A^{5527}$ | $A^{5523}$^$A^{5524}$ |
| ITS1 | E | $^{5605/5608}$ $G^{5606}$/$G^{5609}$ ~$G^{5551}$ | $C^{5587}$^$A^{5589}$ ($E^1$) $G^{5600}$^$C^{5601}$ (+78) $G^{5720}$-$A^{5741}$ ($E^2$) |
| | C | -6100-6200 6117-6192 $C^{6162}$/$C^{6177}$ | $A^{6131}$^$AGA$^$A^{6134}$ $A^{6147}$-$C^{6153}$ |
| | 2 | 6477/6482(8) -6470(9) $C^{6469}$-$C^{6476}$ | $C^{6376}$-$G^{6381}$ ($2^1$) $A^{6458}$^$G^{6459}$ ($2^2$) |
| ITS1 (5′ of 5.8S) | B1 | 6617,6623 6618 $C^{6617}$-$A^{6622}$ | $G^{6592}$-$G^{6602}$ $A^{6600}$^$C^{6601}$ |
| ITS2 (3′ of 5.8S) | 4′ | 6779 | $C^{6755}$-$U^{6757}$ |
| ITS2 | 4a | 6950/6980 ~$C^{6947}$ | $C^{6861}$^$C^{6862}$ ($4a^1$) |
| | 4 | -7600? -7570? | $C^{7559}$^$GCGGTGT$^$C^{7566}$ |
| ITS2 (5′ of 28S) | 3′ | 7935 | $A^{7921}$-$C^{7925}$ |
| 3′ETS (3′ of 28S) | O2 | 12969 | $U^{12985}$^$UUGU$^$C^{12990}$ |

Processing sites from three reviews[8–10] were compared with those identified in this study. Cleavage sites are marked by (^), and regions spanning nucleotide sites are indicated by (-). It's important to note that rRNA annotation can vary positionally due to sequence heterogeneity or incomplete rRNA annotation. In this work, all results are based on the 45SN1 sequence (GeneID:106631777; see "Materials and methods"), ensuring consistency with these specific annotations.

demonstrated[39]. Ribosomal protein RPL3, which binds near the end of 28S rRNA but whose role in 3′ ETS processing had not been established, was also tested.

First, we assessed the depletion of each factor using northern blots, confirming the expected precursor accumulation (Fig. 4B and Fig. S10–15). For NanoRibolyzer analysis, we chose the optimal depletion time point for each factor: 48 h for UTP18, WBSCR22, and LAS1L, and 72 h for DIMT1L, URB1 and RPL3 (Fig. 4B).

RNA from nuclear and cytoplasmic fractions was extracted in triplicate and analyzed using cDNA nanopore-seq, as described above. Coverage profiles revealed distinct differences in processing factor knockdowns compared to controls (Fig. 4C and S16A–C). The effects observed by northern blotting were all confirmed by nanopore sequencing (Fig. 4B–F and Fig. S16).

The depletion of UTP18 led to a substantially increased coverage upstream of site O1, corresponding to the 34S species (Fig. 4C, purple rectangle, and S16B). Knocking down DIMT1L or WBSCR22 increased coverage in ITS1 downstream of the 3′ end of 18S, corresponding to 21S/21S-C (in case of DIMT1L, dark brown rectangle) and 18S-E accumulation (WBSCR22, turquoise rectangle), respectively (Fig. 4C and S16C). Depletion of LAS1L revealed increased reads across ITS2, in agreement with 32S accumulation (Fig. 4C and S16C). Both URB1 and RPL3-depleted samples showed increased coverage across the 3′ ETS (Fig. 4C and S16A). For URB1, this confirms the accumulation

of 3′ ETS extended species in URB1-deficient cells[39]. The observation upon RPL3 knockdown reveals it is also indispensable for this maturation step. Each RNA species was quantified and visualized as heatmaps (Fig. 4D and S17A, S18B, C) or histograms (Fig. S17B–E), confirming the observations on the IGV traces (Source Data 1).

Upon instructing the supervised approach with detailed maturation steps of 18S rRNA 3′ end (see above), it was possible to zoom in further. Notably, it was possible to see that all nuclear precursors of 18S-E pre-rRNA accumulate upon WBSCR22 knockdown (Fig. S18D–F), in full agreement with ref. 36.

In addition to confirming processing phenotypes previously associated with the selected factors, the nanopore-seq approach revealed several unexpected features. For example, we observed a striking and highly reproducible accumulation of fragments encompassing part of the 5′ ETS starting at site O1 upon URB1 and RPL3 depletion (see green and dark yellow rectangles in Fig. 4C and S16B). We also detected a remarkable "processing fingerprint" consisting of multiple consecutive cleavages within the 3′ end regions of the 18S and 28S sequences, which we interpret as increased turnover (Fig. 4E and S16F, Source Data 1). This is particularly interesting given that both factors are important for the same maturation step: removal of the 3′ ETS (Fig. 4C and S16A). Just as unexpected, we observed that DIMT1L, a small-subunit assembly factor[37], also affects 3′ ETS processing and 32S maturation (Fig. 4C, D and Fig. S16), a phenotype that was not apparent by northern blotting.

Lastly, using the unsupervised approach, we mapped the nuclear reads onto an intensity matrix (Fig. 4F and S19A–F). Upon UTP18 depletion, a unique hub corresponding to 34S was detected (purple arrow, Fig. 4F and S19A). Upon DIMT1L depletion, hubs associated with 21S/21S-C and 26S were seen (brown arrows), indicating impaired processing between sites C and 2, in combination with defects in the 5′ ETS (Fig. 4F and S19B). Additionally, we observed notable accumulation of early precursors such as 47S and 45S, consistent with previous findings[37]. Depletion of WBSCR22 revealed accumulation of a hub associated with 18S-E (turquoise arrow), with shortened start coordinates, indicating some level of 5′–3′ degradation of 18S precursors (Fig. 4F and S19C). Knockdown of LAS1L showed reduced aggregated reads at processing site 4 and accumulation at 32S (orange arrow, Fig. S19D). URB1 and RPL3 knockdown exhibited accumulation of reads downstream of processing site O2 (3′ end of 28S) corresponding to RNAs that have retained the 3′ ETS (green arrows, Fig. 4F and S19E, F). Additionally, upon depletion of URB1 and RPL3, we observed an increased accumulation of products upstream of the O1 site, consistent with the coverage profiles (S19E, F), and an increased signal across mature rRNA regions (S19E, F), reflecting by-product buildup and elevated turnover (Fig. 4E and S17F).

In summary, NanoRibolyzer captures previously described pre-rRNAs, newly identified pre-rRNA species, and mature rRNAs; it quantifies them, maps processing sites at single-nucleotide resolution, confirms previously assigned functions of processing factors, reveals new involvements at additional steps, and ultimately defines novel unique processing "fingerprints".

### NanoRibolyzer uncovers spatio-temporal dynamics of RNA modification

Ribosomal RNA is abundantly decorated by covalent modifications[36,40]. To investigate the spatial dynamics of these modifications, we employed direct RNA sequencing (DRS), a technique that sequences native RNA molecules and detects modifications through distinct current signatures that differentiate modified from unmodified nucleotides[41].

First, we selected three base modifications known to occur during late 18S biogenesis: $N^7$-methylguanosine m⁷G, deposited by WBSCR22 at G1639 (refs. 36,37), $N^6,N^6$-dimethyladenosine ($m_2^6Am_2^6A$) deposited by DIMT1L at A1735 and A1736 (ref. 37) and m¹acp³Ψ which is finalized

by addition of the acp group by TSR3 at U1248 (Ref. 42) (Fig. S20A–C). To obtain ground truth, we generated an in vitro transcribed sample (IVT 18S) to serve as a constitutively unmodified negative control. Additionally, we used RNA extracted from cells inactivated for DIMT1L, WBSCR22, or TSR3 to serve as negative controls. We performed signal mapping refinement (conversion of sequence to raw signal) using *Remora* and analyzed the deviations around each modification site across late precursors (21S, 21S-C, 18S-E) and mature rRNA (18S) in both nuclear and cytoplasmic RNA.

For all three modifications, the signal corresponding to precursors closely resembled that observed in inactive mutants or IVT controls, indicating a lack of modification at these sites (Fig. S20D–F). In contrast, cytoplasmic 18S showed a distinct signal shift at these positions, consistent with the presence of the modification on mature rRNA. For m⁷G, the shift between unmodified (IVT, inactive enzyme, and precursors) and mature cytoplasmic RNA was obvious (Fig. S20D). For $m_2^6Am_2^6A$, it was less obvious but visible (Fig. S20E). For m¹acp³Ψ, the situation was a bit more complex to interpret because this modification is formed in three steps: sugar 2′-O methylation by SNORA13, base methylation by EMG1, and addition of acp by TSR3. In Fig. 1C, we discussed that m¹acp³Ψ is readily detected by nanopore sequencing in the cytoplasm but not in the nucleus because the addition of the bulky acp moiety is a late event. Here the IVT control lacking all three modifications (yellow trace), and the sample extracted from cells inactivated for TSR3 lacking only the acp group (black trace) are clearly shifted from each other and from the fully modified mature cytoplasmic 18S rRNA (in red). The nuclear precursor traces (21S, 21S-C, 18S-E) superimpose nicely with the sample inactivated for TSR3, confirming the earlier prediction that the addition of the acp moiety is a late event.

In conclusion, the NanoRibolyzer pipeline can assign nucleotide modification status to a particular pre-rRNA precursor based on raw signal analysis.

Next, we aimed to detect the abundant pseudouridine (Ψ), a uridine isomer formed by DKC1 guided by box H/ACA snoRNAs[43]. For this, we first used our nuclear and cytoplasmic RNA fractions and employed SeqTagger[44] to multiplex up to four DRS barcodes per flow cell (two replicates per condition). Following sequencing, pseudouridine levels were quantified using *Dorado* (v7.2.0) with the "PseU basecaller". In addition to the Ψ ratio provided by the basecaller, we calculated the U-to-C mismatch ratio, as previous studies have shown that isomerization of U to Ψ may lead to U-to-C mismatches during base calling[41,45,46], a feature not integrated in the "PseU basecaller" (see Supplementary Note S1). To ensure selectivity, we filtered out sites with a modification ratio below 10% in each condition, and we used IVT 18S and 28S as negative controls.

We reproducibly detected all known 104 Ψ sites in 18S, 5.8S, and 28S rRNAs, as previously identified by mass spectrometry[40] (Fig. S20G and Supplementary Data 1). We observed that not all positions are fully modified, confirming the existence of compositionally different ribosomes with potentially specialized functions in cells. We conclude that NanoRibolyzer, coupled with "PseU basecaller", is a robust tool for detecting ribosomal RNA pseudouridines. Using it in combination with C/U mismatch detection proved useful as it allowed to disregard hits at positions not known to be modified, such as in the pre-rRNA spacers (Fig. S20G, see e.g., green peaks in the 5′ ETS).

## NanoRibolyzer captures cotranscriptional folding of the 5′ external transcribed spacer

Our cDNA sequencing data revealed a consistent ~1,916-nt deletion within the 5′ ETS, a feature that was absent from the corresponding DRS dataset (Fig. S21A). When we mapped this deletion onto a secondary-structure model of the 5′ ETS generated with *RNAstructure*[47], we realized that it corresponds to a reverse-transcriptase bypass of a strongly predicted structural element,

characterized by very extensive Watson-Crick base-pairing and a notably low ΔG, flanked by cleavage sites A0 and 1 (Fig. S21B, C). The extended folding of the 5′ ETS, as noted by pioneers[48], brings the distant A0 and 1 sites into close proximity, facilitating coordinated processing (Fig. S21C).

## Aberrant RNA precursors are hypomodified

Lastly, to gain information about the spatio-temporal conversion of U in Ψ, we followed the "pre-rRNA sequential extraction" (PSE) method[49], which in our hands led to the isolation of a cytoplasmic (Cp), a nucleoplasmic (Np), and a nucleolar (No) fraction, as confirmed by Western blot analysis with compartment-specific markers (Fig. 5A and S22A).

We sequenced cDNA in triplicate and examined coverage and precursor abundance across the subcellular fractions (Fig. S22B–D). The nucleolar fraction (No) showed strong enrichment for transcripts spanning the 5′ and 3′ ETS and the ITS1 and ITS2 regions (Fig. S22B). In contrast, transcripts in the cytoplasmic fraction (Cp) were dominated by mature rRNAs. Transcripts enriched in the nucleoplasm (Np) displayed intermediate characteristics, having already undergone partial maturation (Fig. S22B). Formal quantification revealed that the nucleolar fractions were enriched in the majority of pre-rRNA species, including 30S, 26S, 21S/21S-C, as well as 34S and 36S-C precursors, considered short-lived aberrant species[6] (Fig. S22C, D, Source Data 1). Progressive processing was reflected by a gradient of precursor abundance, with the highest levels in the nucleolus, followed by the nucleoplasm and then the cytoplasm.

We next performed direct RNA sequencing (DRS) on these fractions to assess the dynamic conversion of U to Ψ from the nucleolus to the cytoplasm. Using the supervised approach revealed highly similar precursor abundance profiles between the cDNA and DRS datasets (Fig. 5B and Fig. S22E, F). Using the unsupervised approach, we identified in the DRS data, processing hubs corresponding to all key precursors (e.g., 30S, 26S, 21S, etc., see arrowheads in Fig. 5C). In addition, we could now also detect in the nucleolar fraction the primary transcript, 47S, and its immediate derivative, 45S (Fig. 5C and Fig. S22G).

Interestingly, this led us to discover novel early intermediates, which we tentatively named 47S-01 (lacking the sequence upstream of the 01 site) and 47S-02 (without the downstream sequence of the 02 site) (Fig. 5C and Fig. 6). These intermediates were already visible in the processing perturbation intensity matrices, particularly upon DIMT1L depletion (Fig. 4F). We now have further proof by direct RNA sequencing that they do exist, as these previously undocumented species presumably escaped detection due to the lower sensitivity and resolution of the methods available at the time.

We conclude that, compared with cDNA sequencing, DRS provides higher resolution, reduced background, and improved capture of longer precursors (Fig. S22G, H). We attribute these differences to the library-preparation workflow and to the use of a recently released, more processive enzyme (see Supplementary Note S2). Despite these distinctions, precursor abundances measured by cDNA sequencing and by DRS were remarkably similar (Fig. 5B and Fig. S22E, F).

Lastly, using the *PseU* basecaller, we reproducibly identified all expected Ψ sites in nucleolar, nucleoplasmic, and cytoplasmic fractions (Fig. S23 and S24; Supplementary Data 1). Having categorized the reads according to their respective precursor sequences (using the supervised approach), we next performed precursor-specific pseudouridine analysis to investigate the modification dynamics across the different precursors and cell compartments. A key conclusion from the precursor-specific modification mapping is that Ψ sites are already largely present on the primary 47S transcript. Another important observation is that aberrant RNAs, such as the 34S and 36S-C species, are distinctly hypomodified (Fig. 5D, E). At this stage, we cannot determine whether these RNAs are hypomodified because they are

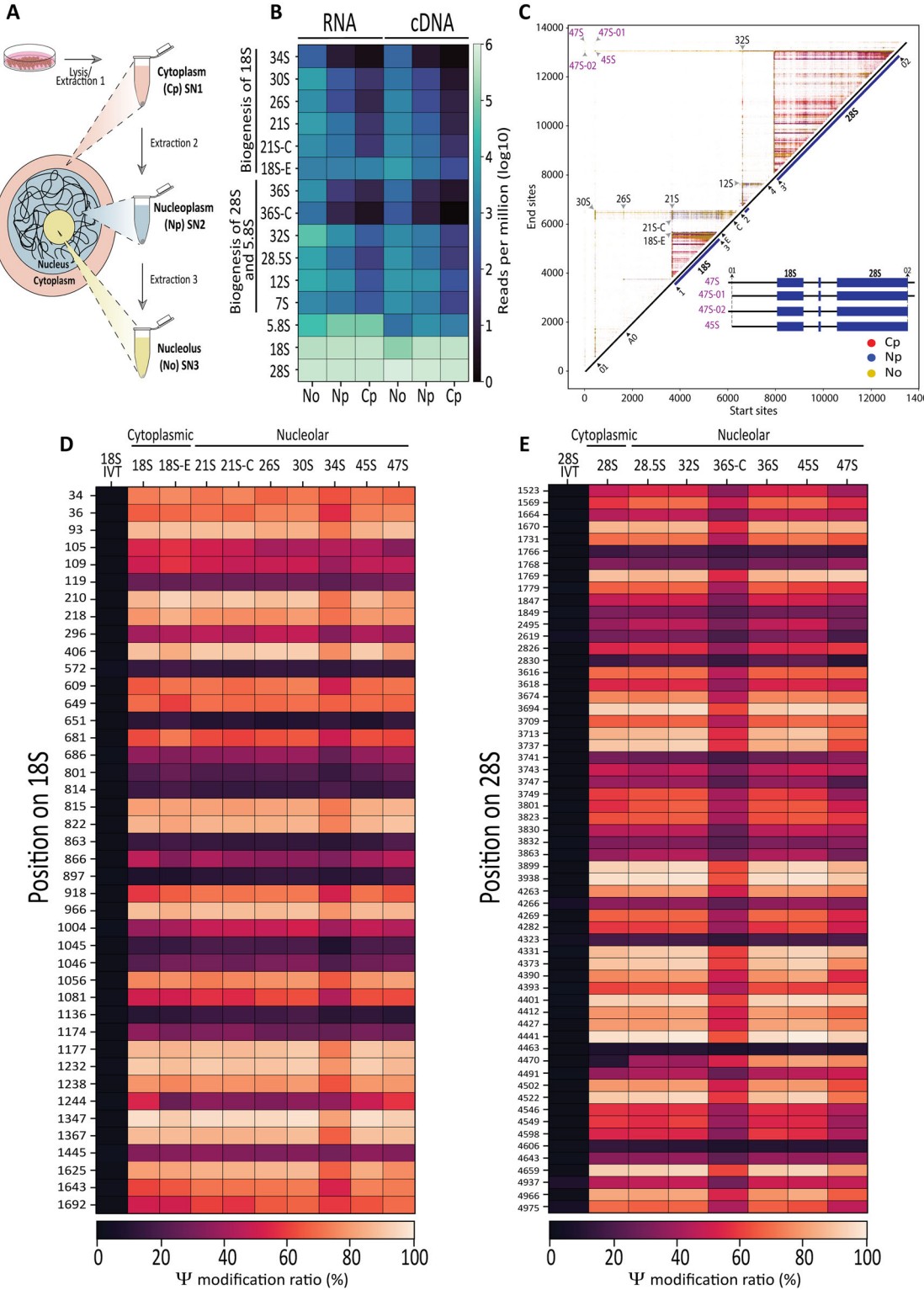

**Fig. 5 | Spatio-temporal mapping of precursor-specific RNA modifications.**
**A** Schematic representation of the Pre-rRNA Sequential Extraction (PSE) method developed by the team of Prof. Dosil[49]. The SN1, SN2 and SN3 fractions are enriched for the cytoplasm (Cp), nucleoplasm (Np) and nucleolus (No) fractions, respectively (see analysis of marker proteins by western blots in Fig. S21A). **B** Quantification of detected pre-rRNA intermediates and mature rRNAs in the nucleolus, nucleoplasm and cytoplasm ($n = 3$ samples each) using direct RNA sequencing (RNA, left) and cDNA sequencing (right). The results are remarkably similar.
**C** Overlayed intensity matrices of nucleolar (yellow), nucleoplasmic (blue), and cytoplasmic (red) reads. Key intensity hubs and their associated precursors are marked with arrows. The hubs detected at the top left of the 2-D display (in purple) correspond to the primary transcript, the 47S, and its derivatives: the 47S-01 and 47S-02 (both newly detected species) and the 45S. A schematics in the bottom right corner of the display illustrate the structure of the 47S and of its direct derivatives.
**D** Heatmap depicting the level of modification of each of the 42 pseudouridines detected on 18S and mapped to each individual precursor. The nucleolar signal was used to quantify Ψ in 18S-E, 21S-C, 21S, 26S, 30S, 45S, and 47S. For quantifying Ψ in 18S, it is the cytoplasmic signal that was used. **E** As in panel (**D**) for the 60 sites present in 28S, in vitro transcribed 18S and 28S were used as negative controls.

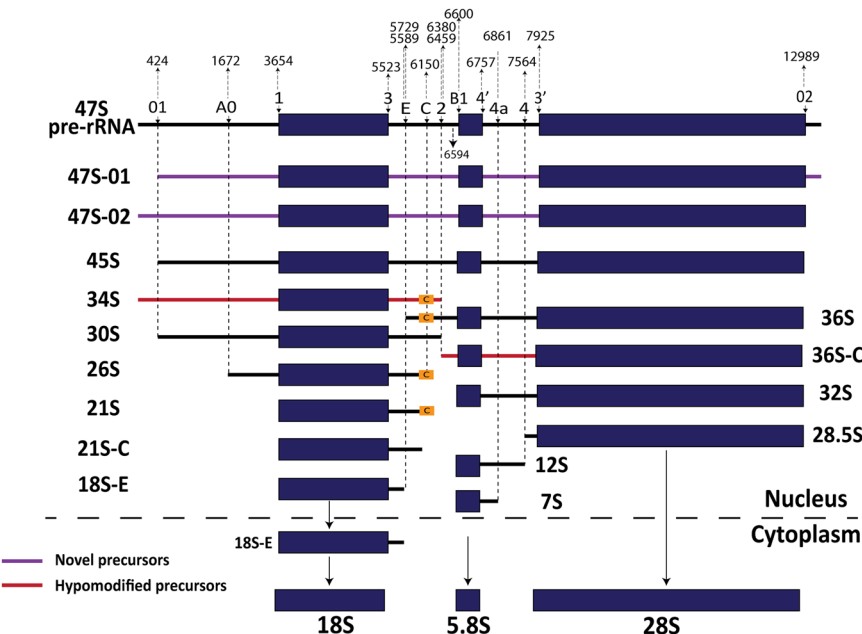

**Fig. 6 | Current model of human pre-rRNA processing and redefined processing sites.** Novel 47S derivatives (47S-01 and 47S-02, in purple) and hypomodified precursors (34S and 36S-C, in red) are shown. All processing sites defined to single-nucleotide resolution are indicated. See Fig. S6 for more details.

aberrant, or whether insufficient modification contributes to their aberrant nature.

In conclusion, NanoRibolyzer can map rRNA modification at the individual precursor level.

## Discussion

Ribosome biogenesis is a sophisticated pathway involving hundreds of interconnected steps[1,2]. Among these, pre-rRNA processing, i.e., RNA cleavage to generate mature rRNA ends, and pre-rRNA modification serve as excellent *proxies* for monitoring the overall process (Fig. 1A).

Traditionally, pre-rRNA processing intermediates have been analyzed using metabolic labeling, northern blotting, or primer extension. While these techniques are robust, they are limited in resolution, sensitivity, and throughput, and often require access to costly and hazardous radioactive materials. Moreover, they typically detect only a few dozen abundant precursors, despite the well-established existence of thousands of intermediates generated by progressive, one-nucleotide-at-a-time digestion by exoribonucleases, particularly during 3′ end maturation of the 18S and 5.8S rRNAs[50]. This extraordinarily rich population of low-abundance, metastable intermediates has remained largely inaccessible until now. NanoRibolyzer was designed to address this limitation.

RNA modification represents a second key dimension of RNA metabolism. While the diversity and distribution of rRNA modifications have been well established on mature rRNAs[51], equivalent information at the level of individual processing intermediates has been lacking. NanoRibolyzer also addresses this limitation.

NanoRibolyzer is a long-read, nanopore sequencing-based pipeline for the comprehensive detection and quantification of pre-rRNA intermediates, coupled with the concomitant mapping of RNA modifications.

To gain spatio-temporal resolution of pre-rRNA processing and modification, NanoRibolyzer was applied to purified subcellular fractions. In this study, sequencing was primarily performed on nuclear and cytoplasmic enriched fractions (Fig. 1B). A straightforward, high-throughput-compatible protocol was purposefully developed (Fig. 1B); this was a key prerequisite for successfully establishing the method, as abundant and stable mature rRNAs would otherwise mask the

detection of low-abundance, transient nuclear intermediates. Ultimately, we extended the method to include nucleolar, nucleoplasmic, and cytoplasmic fractions (Fig. 5A). One key conclusion was that the cytoplasm, as expected, contains almost exclusively mature rRNAs, while the nucleolus and nucleoplasm are highly enriched in a diverse population of precursors retaining variable lengths of spacer sequences. In the cytoplasm, we could also monitor final trimming events of 18S rRNA precursors (Figs. S8 and S9).

A key feature of NanoRibolyzer is that it integrates two complementary mapping strategies: (i) a supervised approach, built on prior knowledge and the large body of literature on pre-rRNA processing, and (ii) an unsupervised approach, which remains entirely agnostic of existing annotations.

The unsupervised or discovery mode was particularly powerful to appreciate the diversity and richness of the reads, which we plotted in a two-dimensional matrix based on their starting and ending coordinates. This output provides an intuitive visualization of processing efficiency under different conditions (e.g., normal *vs.* perturbed/disease states). In these 2D matrix displays, we could visualize exoribonucleases "in action" digesting pre-rRNAs one nucleotide at a time. RNA ends were perfectly aligned on horizontal or vertical lines in the case of 5′→3′ and 3′→5′ digestion, respectively (Fig. S6), and the sites of exoRNase pausing became evident.

By combining the two mapping approaches, we validated all known processing sites, often refining them to single-nucleotide resolution, and identified novel ones (Table 1, Fig. 3, and S7). We also discovered previously uncharacterized pre-rRNA species, notably novel derivatives of the primary transcript (47S-01 and 47S-02), suggesting some plasticity at least in the sequence of the initial cleavage events (Fig. 6).

We benchmarked our mapping strategy through high-resolution re-analysis of depletion of seven known processing factors (Fig. 4). These factors were selected to span maturation across all four spacer regions. NanoRibolyzer not only recapitulated their known functions but also uncovered previously unappreciated roles. For example, DIMT1L, classically implicated in ITS1 processing, was also found to affect the 5′ ETS and 3′ ETS. This highlights that NanoRibolyzer will become an essential tool for future ribosome-biogenesis research.

Throughout our analyses, we identified an underappreciated diversity of rRNA-derived fragments, spanning both non-coding spacers and mature rRNAs. Some of these fragments likely carry biologically relevant information, as suggested by the highly similar patterns observed upon depletion of factors acting at the same step of ribosome biogenesis. A notable example is provided by URB1 and RPL3, whose depletion impairs removal of the 3′ ETS (Fig. 4), and leads to the accumulation of fragments derived from the 5′ ETS (just downstream of site 01), as well as from the 3′ ends of 18S and 28S rRNAs (Fig. 4E). Further studies will determine the extent to which such molecular "fingerprints" may serve as markers of specific perturbations or disease states.

We used both long-read cDNA sequencing (suited for processing studies and, at the time, the only protocol supporting multiplexing), and direct RNA sequencing (DRS), which allows RNA modification detection and RNA end characterization. For mapping and quantifying pre-rRNA and mature rRNA species, results from both approaches were remarkably consistent (Fig. S22).

By design, NanoRibolyzer, like all sequencing-based approaches, entails inherent selection steps arising from cDNA library preparation and nanopore biophysics; by contrast, Northern blotting is a largely "blind" technique in which RNAs are directly resolved without prior selection, and therefore remains a valuable and complementary method.

NanoRibolyzer also maps RNA modifications, as demonstrated here with the detection of pseudouridines, known to occur at 104 sites on human rRNAs, and several base modifications (e.g., $m_2^6Am_2^6A$, $m^7G$, $m^1acp^3\Psi$). On mature rRNAs, we confirmed all pseudouridine positions previously mapped by RNA mass spectrometry[40]. By analyzing pseudouridines on individual precursor rRNAs, we found that many of the modifications present on mature rRNAs were already detectable on the primary transcript (47S), indicating early modification events.

We also observed that numerous positions were only substoichiometrically modified, confirming previous reports and supporting the concept of ribosome heterogeneity[52]. Most notably, aberrant pre-rRNAs involved in both small and large subunit biogenesis, such as 34S and 36S-C species (Fig. 5D-E), were markedly hypomodified. Whether this hypomodification is a cause or consequence of the aberrant nature of these RNA species remains to be determined.

Unexpectedly, we also noted a systematic -two-kilobase deletion in our long-read cDNA data (Fig. S20), which, upon closer inspection, corresponded to a reverse transcriptase template switch across a predicted strong secondary structure. This region has previously been proposed to spatially coordinate processing sites A0 and 1 (ref. 48).

A few recent studies have begun to use long-read sequencing (Oxford Nanopore Technologies, PacBio HiFi) to explore distinct aspects of rRNA biology, including rRNA heterogeneity (SNPs)[53,54], rRNA modification[17,55–57], and rRNA processing[17,58], in archaea, yeasts, and human cells. Compared to protocols that rely on circularization or affinity purification of precursor ribosomes (which may introduce bias) and sequencing of associated RNAs, NanoRibolyzer is simple, robust, and easy to implement.

With the continued evolution of nanopore sequencing technology (e.g., the recent availability of multiplexing in DRS), NanoRibolyzer is poised to evolve further. In the near future, we foresee extending its use to define rRNA subtypes based on patterns of co-occurring or mutually exclusive RNA modifications and to assess the impact of SNP variants on RNA modification.

In summary, NanoRibolyzer provides a simple, versatile, nanopore sequencing-based platform to study pre-rRNA processing and modification in human cells and, in the future, in other systems. It overcomes key limitations of classical RNA biochemical methods. By enabling single-nucleotide resolution and quantitative analysis of pre-rRNA intermediates, together with RNA modification mapping at the level of individual species, it offers invaluable insights into ribosome biogenesis, with broad applications in basic research and in the study of disease mechanisms, including ribosomopathies and tumorigenesis, as well as in clinical diagnostics.

## Methods

### Cell lines and culture
HEK293 cells (ATCC CRL-1573) were cultured in DMEM supplemented with 10% FBS and 1% L-glutamine and maintained in an incubator at 37 °C and 5% $CO_2$.

All mutation analyses, including tsr3 $-/-$[42], dimt1l-Y131G[55] and wbsr22-D82K[55], were introduced in homozygous diploid HCT116 p53-positive cells.

### siRNA inactivation experiments
Cells were revere transfected with silencers (10 nM, except URB1: 15 nM) in a time course (H6, H12, H24, H48, H72) to identify the best condition for Nanoribolyzer analysis[6]. All DsiRNAs silencers were used at 10 nM final (except for URB1, 15 nM). Silencers against UTP18, WBSCR22, DIMT1L, and LAS1L are Silencer Select (Ambion). Silencers against URB1, RPL3, and RPS26 are DsiRNAs (IDT). The Silencer RNA sequences are shown in Table S2.

### Simplified nuclei isolation protocol
Detailed description of the protocol is illustrated in the Supplementary Fig. S1. Briefly, samples were trypsined and washed with cold PBS, after which they were resuspended in Nuclei Isolation Buffer (NIB − 10 mM Tris-HCl (pH 7.4),10 mM NaCl, 3 mM $MgCl_2$, 0.1% Igepal, 0.1% Tween-20,1% BSA, 0.15 mM Spermine, 0.15 mM Spermidine, 0.2 U/μl RNase inhibitor) and homogenized using a loose pestle with ten strokes. The homogenate was then incubated on ice for 15 min and subsequently centrifuged at $100 \times g$ for 5 min at 4 °C. The resulting soluble fraction (cytoplasmic) was transferred to a new tube, and Trizol was added to the sample at room temperature (RT) while nuclei isolation continued. The remaining pellet was subjected to two washes with 500 μl of ice-cold NIB buffer and centrifuged at 100 x g for 5 min at 4 °C. After the second wash, the supernatant was removed, and the pellet was resuspended with 300 μl of ice-cold NIB buffer. For nuclei isolation, an equal volume (300μl) of 50% Optiprep solution (Stemcell technologies, 07820) was added to the homogenate sample and thoroughly resuspended by pipetting, resulting in a 25% sample/Optiprep mix. Then, 600 μl of 40% Optiprep solution, followed by 30% Optiprep solution, were layered in a 2 ml Eppendorf tube. The 25% sample/Optiprep mix was layered on top of the 30% solution, forming three visible layers. The tube was centrifuged at top speed ($20,817 \times g$) in a bench centrifuge for 20 min at 4 °C. After centrifugation, the nuclei (~600 μl) were carefully collected from the 40%-30% phase and transferred into a 1.5 ml Lo-bind tube. 600 μl of Nuclei Wash Buffer (NWB−10 mM Tris-HCl (pH7.4), 10 mM NaCl, 3 mM $MgCl_2$, 0.1% Tween-20,1% BSA, 0.2 U/μl RNase inhibitor) was added to the nuclei and thoroughly resuspended by pipetting. The sample was then centrifuged at $150 \times g$ for 10 min at 4 °C, and the supernatant was removed. The nuclei were washed again with 500 μl of ice-cold NWB buffer and centrifuged similarly. The supernatant was removed, leaving approximately 20 μl of solution. Quality of the nuclei was assessed using DAPI staining and visualized under the microscope using a UV filter to identify DAPI-positive nuclei. High-quality nuclei were characterized by debris-free, round or oval-shaped DAPI-stained nuclei. Once the quality of isolated nuclei was confirmed, the remaining nuclear samples were treated with DNase I (NEB). The DNase I master mix was supplemented with 0.15 mM spermidine, 0.15 mM spermine, and RNAse inhibitor (U/ul) per sample. The samples were then incubated at 37 °C for 20 min. Next, the sample volume was brought to 500 μl using nuclease-free water, and 500 μl of TRIzol reagent (Life Technologies, 15596026) was added to the nuclear samples to proceed with RNA isolation.

## Pre-ribosomal sequential extraction method (PSE)

The PSE method was performed as previously described[49]. Briefly, cells (HEK293) were grown at ~80% confluency and harvested on ice-cold phosphate-buffered saline. Cells were resuspended thoroughly in 0.5 ml of SN1 buffer (20mM HEPES-NaOH [pH 7.5], 130 mM KCl, 10 mM MgCl$_2$, 0.05% Igepal CA-630), supplemented with Cømplete protease inhibitor cocktail (Roche), followed by centrifugation at 1300 × $g$ for 3 min at 4 °C. The supernatant was collected and stored as the SN1 fraction. The pellet was washed with 0.5 ml SN1 buffer, and then resuspended in 0.3 ml of SN2 buffer (10 mM HEPES-NaOH (pH 7.5), 10 mM NaCl, 5 mM MgCl2, 0.1% Igepal CA-630, 0.5 mg/ml heparin, 600 U/ml RNasin (Promega)) supplemented with 100 U RNase-free DNase I (Qiagen), and incubated for 10 min at room temperature with gentle mixing. The lysate was centrifuged 12,300 × $g$ for 10 min at 4 °C, and the supernatant was collected as the SN2 fraction. The remaining pellet was resuspended in 0.4 ml of SN3 buffer (20 mM HEPES-NaOH (pH 7.5), 200 mM NaCl, 4 mM EDTA, 0.1% Igepal CA-630, 0.04% sodium deoxycholate, 4 mM imidazole, 0.1 mg/ml heparin, 1 mM dithiothreitol (DTT), Cømplete, 600 U/ml RNasin) and incubated for 20 min at room temperature with moderate agitation. The extract was centrifuged 12,300 × $g$ for 10 min in 4 °C, and the supernatant was collected as the SN3 fraction. Total RNA was prepared from each fraction using TRIzol reagent (Life Technologies, 15596026) according to the manufacturer's protocol.

## RNA isolation

RNA isolation was performed as previously described[59]. Nuclear and cytoplasmic (as well as whole cell) fractions (~1 ml) were incubated at RT for at least 5 Min. 200 μl Chloroform was added to nuclear and cytoplasmic fractions. Samples were Vortexed for 15 s, incubated at RT for 3 min and centrifuged for 15 min full speed (FS) at 4 °C. The upper aqueous phase (~550 μl) was transferred into a fresh Eppendorf tube and 500 μl isopropanol was added, thoroughly resuspended and incubated at RT for 15 min. Samples were centrifuged for 10 min at full speed at 4 °C. The supernatant was discarded pellet was washed with 75% EtOH in nuclease-free water, followed by a 5 min centrifugation at 7500 × $g$ at 4 °C. The supernatant was discarded, and the pellet was air-dried for 5-10 min. RNA was eluted with 30μl of RNAse-free water and mixed in a Hula mixer for 10 min at RT. RNA concentration was measured with qubit and RNA integrity was assessed via Agilent RNA ScreenTape analysis.

## Northern blotting

Total RNA was extracted using TRIzol reagent, separated on agarose denaturing gel, and analyzed by northern blotting, as previously described[6]. The depletion optimization was performed on HEK293 cells, where each depletion factor displayed the expected phenotype according to the literature. On the basis of this analysis, the following depletion time points were selected for NanoRibolyzer: UTP18 (48 h), DIMT1L (72 h), WBSCR22 (48 h), LAS1L (48 h), and URB1 (72 h) (see Fig. 3b). The probes used for Northern blotting are shown in Table S3.

## Western blot controls for SN1, SN2, and SN3 fractionation

Equivalent amounts of SN1, SN2, and SN3 fractions, each extracted from starting material equivalent to 25 μg of total protein (fractionated according to ref. 49), along with 25 μg of total protein from whole-cell lysate used as a control, were separated on a 10% SDS-PAGE gel. Proteins were transferred to nitrocellulose membranes and probed in TBS containing 0.1% Tween-20 and 3% BSA. Membranes were incubated with either: (1) Anti-rabbit primary antibodies: anti-histone H3 (Proteintech, Ref 17168-1-AP), anti-nucleolin (Proteintech, Ref 10556-1-AP), and anti-α-tubulin (Proteintech, Ref 11224-1-AP), each at 1:5000 for 2 h, followed by HRP-conjugated anti-rabbit secondary antibody (Cytiva, Ref NA934V) at 1:5000 for 2 h, or (2) Anti-mouse primary antibodies: anti-fibrillarin (Antibodies-online, Ref ABIN361375) or anti-UBF (Tebu

Bio, Ref SC-13125), each at 1:1000 overnight, followed by HRP-conjugated anti-mouse secondary antibody (Jackson ImmunoResearch, Ref 115-035-062) at 1:5000 for 2 h.

## In vitro polyadenylation using poly(A) tailing of RNA

For in vitro polyadenylation, the "In vitro poly adenylation using poly(A) tailing of RNA" kit was used (NEB#M0276), according to the manufacturer's instructions. Briefly, 1 μg of RNA was taken in 15 μl nuclease free water and supplemented with 2 μl of 10× *E. coli* Poly(A) Polymerase Reaction Buffer, 2 μl ATP (10 mM) and 1 μl *E. coli* Poly(A) Polymerase (to a total volume 20 μl). Samples were incubated at 37 °C for 30 min. The polyadenylated RNA samples were purified using RNAClean XP beads (Beckman Coulter, A63987), according to the manufacturer's instructions. In the last elution step, the sample was resuspended in 10 μl nuclease-free water and incubated at 37 °C for 5 min in a Hula mixer. The sample was placed on the magnet, and once the solution was clear, the elute was transferred into a clean 1.5 ml Eppendorf tube.

## In vitro transcription of 18S and 28S rRNA

Synthesis of the 18S and 28S rRNA in vitro transcripts was performed using HiScribe® T7 High Yield RNA Synthesis Kit following the manufacturer's instructions. Briefly, 2 μg linearized plasmid containing the mature rRNA (GenScript) was used as template material, combined with 10× Reaction Buffer, 10 mM NTPs and 2U T7 RNA Polymerase Mix. Reactions were incubated for 2 h at 37 °C and stopped by digestion of the template plasmid through DNase I (ThermoFisher Scientific, EN0525) following manufactures instructions. Purification was performed using Monarch® RNA Cleanup Kit (NEB, T2040), and product quality was assessed using capillary electrophoresis via Agilent RNA ScreenTape Analysis.

## Direct cDNA-native barcoding library preparation

Direct cDNA coupled with native barcoding libraries was prepared using the Direct cDNA Sequencing Kit (SQK-DCS109), Native Barcoding Expansion 1-24 (EXP-NBD104, EXP-NBD114), following the manufacturer's protocol. Reverse transcription and strand-switching. 1 μg of poly(A)-tailed RNA was transferred to a 1.5 ml tube and adjusted to 7.5 μl with nuclease-free water. In a 0.2 ml PCR tube, 7.5 μl of RNA sample were mixed with 2.5 μl of VNP (ONT), 2.5 μl of 10 mM dNTPs (NEB N0447), and the volume was adjusted to 11 μl with nuclease-free water. The samples were incubated for 10 min at room temperature and then snap-cooled on a pre-chilled freezer block for 1 min. Next, a master mix was prepared, containing 4 μl of 5x RT Buffer (ThermoFisher, EP0751), 1 μl RNaseOUT (Life Technologies, 10777019), 1 μl of Nuclease-free water, and 2 μl Strand-Switching Primer (SSP, ONT) per sample, to a total volume of 8 μl. The strand-switching buffer was added to the snap-cooled, annealed mRNA, and the samples were incubated at 42 °C for 2 min in the thermal cycler. Subsequently, 1 μl of Maxima H Minus Reverse Transcriptase (ThermoFisher, EP0751) was added, and the total volume became 20 μl. The samples were incubated following a specific thermal protocol: 42 °C for 90 min, 85 °C for 5 min, and then holding at 4 °C. After the reverse transcription, RNA degradation and second-strand synthesis were performed. 1 μl of RNase Cocktail Enzyme Mix (ThermoFisher, AM2286) was added to the reverse transcription reaction and incubated for 10 min at 37 °C. The samples were then subjected to the AMPure XP beads-based (Beckman Coulter A63881) purification method using a 0.85× ratio of beads:sample, and ultimately, cDNA hybrid was eluted in 20 μl of nuclease-free water. Next, the 20 μl of reverse-transcribed samples were prepared with 25 μl of 2× LongAmp Taq Master Mix (NEB, N0447), 2 μl of PR2 Primer (PR2, ONT), and 3 μl of Nuclease-free water, to a total volume of 50 μl. The samples were incubated at specific temperatures in the thermocycler. Afterwards, the samples were then subjected to the AMPure XP beads-based purification method using 0.8× ratio of

beads:sample and ultimately cDNA/RNA hybrid was eluted in 21 μl of nuclease-free water. The eluted sample was quantified using a Qubit fluorometer. *End-prep*. Subsequently, end repair and dA-tailing were performed by mixing 20 μl of cDNA sample with 30 μl Nuclease-free water, 7 μl Ultra II End-prep reaction buffer (NEB, E7546), and 3 μl Ultra II End-prep enzyme (NEB, E7546) mix to a total volume of 60 μl. The samples were incubated at 20 °C for 5 min and then at 65 °C for 5 min. Next, the samples were subjected to AMPure XP beads-based purification using 1× ratio of beads:sample, and the cDNA was eluted with 22.5 μl of nuclease-free water. *Barcode ligation*. Barcode ligation was then performed, where 22.5 μl of End-prepped DNA was mixed with 2.5 μl of Native Barcode and 25 μl of Blunt/TA Ligase Master Mix (NEB, M0367) to a total volume of 50 μl. The reaction was incubated for 15 min at room temperature. The samples were then subjected to AMPure XP beads-based purification using 1× ratio of beads:sample, and the cDNA was eluted with 26 μl of nuclease-free water. Lastly, the barcoded samples are pooled to a final volume of 65 μl in a 1.5 ml Eppendorf tube. *Adapter ligation*. Adapter ligation was performed by adding 65 μl of pooled barcoded sample, 5 μl of Adapter Mix II (AMII, ONT), 20 μl of 5X NEBNext Quick Ligation Reaction Buffer (NEB, B6058), and 10 μl of Quick T4 DNA Ligase (NEB, E6056) to a total volume of 100 μl. The final libraries were incubated for 10 min at room temperature and then subjected to AMPure XP beads-based purification, with the cDNA being eluted with 26 μl of nuclease-free water. The sample was loaded and sequenced onto a primed PromethION flow cell as per the manufacturer's instructions.

### Direct RNA library preparation

Direct RNA libraries were prepared using the SQK-RNA004 kit (ONT) following the manufacturer's protocol. Briefly, 1 μg of poly(A)-tailed RNA was adjusted to a final volume of 9.5 μl with nuclease-free water. 3 μl of NEBNext Quick Ligation Reaction Buffer (NEB B6058), 1 μl of RT Adapter (RTA) (ONT), and 1.5 μl of T4 DNA Ligase 2 M U/ml (NEB M0202), were added to the sample, resulting in a total volume of 15 μl. The reaction is mixed by pipetting and incubated for 10 min at room temperature. Next, the reverse transcription master mix was prepared by mixing 9 μl of Nuclease-free water, 2 μl of 10 mM dNTPs (NEB N0447), 8 μl of 5x first-strand buffer (Thermo Fisher Scientific, 18080044), and 4 μl of 0.1 M DTT, resulting in a total volume of 23 μl. The master mix was added to the RNA sample containing the RT adapter-ligated RNA. 2 μl of SuperScript III reverse transcriptase (Thermo Fisher Scientific, 18080044) were added to the reaction, bringing the final volume to 40 μl. The reaction was incubated at 50 °C for 50 min, followed by 70 °C for 10 min, and then brought to 4 °C. Agencourt RNAClean XP beads (Beckman Coulter, A63987) were resuspended and 72 μl of the resuspended beads were added to the reaction. The sample was mixed by pipetting and incubated on a Hula mixer for 5 min at room temperature. Subsequently, the sample was subjected to two washes with 70% ethanol, and the RNA:DNA hybrids were eluted with 20 μl of nuclease-free water. For the adapter ligation reaction, 8.0 μl of NEBNext Quick Ligation Reaction Buffer, 6.0 μl of RNA Ligation Adapter (RLA), 3.0 μl of Nuclease-free water, and 3.0 μl of T4 DNA Ligase were mixed with 20 μl of the eluted sample to reach a total volume of 40 μl. The reaction was incubated for 10 min at room temperature. 30 μl of resuspended RNAClean XP beads were added to the adapter ligation reaction, mixed by pipetting, and incubated on a Hula mixer for 5 min at room temperature. The sample was then subjected to two washes with the Wash Buffer (WSB, ONT) using a magnetic rack. Following the washes, the beads were pelleted on the magnet, and the supernatant was pipetted off. The pellet was resuspended in 33 μl of Elution Buffer (EB, ONT) and was incubated at 37 °C for 10 min in a Hula mixer. Incubation at 37 °C allows the release of long fragments from the beads. The eluate was then cleared by pelleting the beads on a magnet, and the eluate was retained and transferred to a clean to 1.5 ml tube. The sample was loaded and sequenced onto a primed PromethION flow cell as per the manufacturer's instructions.

### Implementation of NanoRibolyzer pipeline

NanoRibolyzer was implemented as a Nextflow-based workflow using Docker containers and could be installed as a plugin within Oxford Nanopore Technologies (ONT) Epi2Me platform (https://github.com/stegiopast/wf-nanoribolyzer). Pod5 output files were basecalled using the Dorado basecaller (https://github.com/nanoporetech/dorado), trimmed with Porechop (https://github.com/rrwick/Porechop) to remove adapter sequences and aligned to the 45SN1 (equivalent to 47S) (GeneID:106631777; NW_021160023.1:480347-493697) using minimap2[60] with the map-ont flag. The read IDs of the aligned 45SN1 were used to filter the original pod5 file. The resulting unaligned BAM files were used to collect metadata on reads at a single nucleotide resolution, and the rebasecalled reads were realigned to the 45SN1 reference. The final BAM files were processed using both template-based and template-free approaches.

NanoRibolyzer is open-source and has been made freely available to the community through the Epi2ME platform for broader accessibility and use.

### Template-based quantification of rRNA precursors

The template-based algorithm associated long-reads with literature-based ribosomal intermediates[8–10]. In this approach, the pairwise minimal reciprocal overlap (MRO) between a query read and all possible intermediates was determined. The MRO was defined by calculating the minimal relative overlap of the query over the intermediate and vice versa. Once the minimal overlap for each query-intermediate pair was established, the pair with the maximal overlap was used to associate the read with the corresponding intermediate (see Fig. S3). Read clusters were then stored in a tab-separated values (TSV) table, which included read IDs, absolute and relative read counts, and the start and end sites of all reads associated with each intermediate. Additionally, bed files for each intermediate were generated to facilitate visualization in the Integrative Genomics Viewer[61] (IGV). The 45SN1 reference FASTA from the NanoRibolyzer references repository was used for all analyses.

### Template-free rRNA precursors

The template-free algorithm was based on the construction of a two-dimensional (length(45SN1)$^2$) intensity matrix in which reads were embedded using the alignment start and end sites as coordinates. The number of reads sharing start and end site coordinates on the matrix led to the formation of intensity "hubs". The resulting intensity matrix was stored in CSV format, which included the start site, end site, number of reads, and ID list for each intensity hub. For visualization of the matrices, absolute read counts of intensity hubs were min-max normalized, applying an additional contrast enhancement of 2%.

### Determination of read cleavage sites

The determination of significantly abundant cleavage sites was computed using alignment start and end sites, where the absolute abundance of start and end sites was determined along the 45SN1 reference. For each template-based intermediate, the mean abundance and standard deviation (SD) of alignment start and end sites were calculated. Cleavage sites occurring two SDs above the mean in the dataset were considered significantly abundant cleavage sites. For overlapping template-based intermediate cleavage site intervals, metrics were determined by calculating the mean of means and the mean of SDs. The output files were stored in TSV and BED file formats, including information about their relative abundance and the cleavage site location. The BED files were visualized using IGV.

## DRS modification analysis

PseudoU detection was performed by using Dorado version 7.2.0 with the super accuracy RNA004 model version 5.0.0 using the "pseU" flag. The resulting unaligned BAM file was converted into fastq format using samtools[62] (https://github.com/samtools/samtools). The fastq file was aligned with minimap2[60] using the "map-ont" long read alignment flag together with the MD flag to ensure the remainder of the modification information. Subsequently, the aligned BAM file was processed using pysam (https://github.com/pysam-developers/pysam). For each possibly modified nucleotide in a read-aligned BAM file, there is a modification probability. A threshold of 95% modification probability was chosen to determine modifications.

For the acquisition of the raw signal analysis at specific modification sites, the data were basecalled and aligned as described above. Template-based analysis information was used to determine intermediate-specific datasets. The aligned BAM file and the original pod5 together with a 45SN1 reference fasta file were processed using Remora (https://github.com/nanoporetech/remora). Remora was used to perform the resquiggleing, which is the process of association of the raw signal interval with specific nucleotides on the reference or basecalled sequence. In this case, once the intervals were associated with the reference sequence, the mean of z-normalized raw current signal covering 10 bases up- and downstream from the loci of interest was extracted. For all reads covering the locus of interest, the mean of means and semi-standard deviation (deviation below and above the mean) of means were determined.

## Statistics

All statistical analyses are described in the respective figure legends. Each legend provides detailed information about the statistical metrics (such as the mean and standard deviation), sample sizes, statistical tests used and any relevant adjustments applied.

## Reporting summary

Further information on research design is available in the Nature Portfolio Reporting Summary linked to this article.

## Data availability

Nanopore sequencing data were deposited in the European Nucleotide Archive (ENA) under project accession number PRJEB82698 [ebi.ac.uk/ena/browser/view/PRJEB82698] and are publicly available as of the date of publication. Owing to repository file size limitations, samples ERS24812595 and ERS24812596 were randomly subsampled before public deposition. If the entire dataset of those samples is required, please contact the authors. All data, including raw numbers for graphs, are available in the Source Data file or in the Supplementary Information. Source data are provided with this paper. Source data are provided with this paper.

## Code availability

All scripts and code used in this work have been made available on GitHub (https://github.com/stegiopast/wf-nanoribolyzer). The preprocessing pipeline for single samples is part of the NextFlow pipeline. Code for downstream analysis is stored in Jupyter Notebooks in the GitHub repository.

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

## Acknowledgements

This work was partly funded by Deutsche Forschungsgemeinschaft (DFG, German Research Foundation); project no. 439669440 TRR319 RMaP, TP A05/C01/C03 to M.H. and S.M, TP B07 to T.B., TP A07 to S.G. and L.L., and TP C04 to S.G and S.P., T.B. and S.G. acknowledge funding from the ReALity initiative of the Johannes Gutenberg University Mainz. Research in the Lab of D.L.J.L. was supported by the Belgian Fonds de la Recherche Scientifique (F.R.S./FNRS), EOS [CD-INFLADIS, grant n°40007512], Région Wallonne (SPW EER) Win4SpinOff [RIBOGENESIS], the COST actions EPITRAN (CA16120) and TRANSLACORE (CA21154), the European Joint Program on Rare Diseases (EJP-RD) RiboEurope and DBAGeneCure. S.G. and L.L acknowledge funding by the Boehringer Ingelheim Stiftung. S.G. acknowledges funding by the M3odel initiative from the Forschungsinitiative Rheinland-Pfalz.

## Author contributions

S.P. developed and implemented the NanoRibolyzer pipeline; S.P. and L.L. analyzed the data. T.B., L.W., and S.P. performed nuclei isolation and RNA isolation experiments. L.W. performed the knockdown experiments and northern blots. S.M. performed the in vitro transcription of 18S and

28S rRNA. T.B. and S.P. prepared and sequenced the nanopore-seq libraries. T.B. and D.L. conceived and supervised the work, with the assistance of S.G., M.H., and B.L., who provided valuable input and feedback in various discussions. T.B., D.L., and S.P. wrote the paper, with contributions from all the authors.

## Funding

## Competing interests

M.H. serves as a consultant for Moderna, Inc. All other authors declare no competing interests.
