## [Transparent Peer Review file · Nature Communications]

Mapping human pre-rRNA processing and modification at single nucleotide resolution using long read Nanopore sequencing

Corresponding Author: Dr Tamer Butto

Version 0:

Reviewer comments:

Reviewer #1

(Remarks to the Author)

This study presents a novel technical approach for quantification and characterization of pre-rRNA intermediates using long-read nanopore sequencing. The authors provide a comprehensive set of validation data showing that their RNA fractionation, sequencing and analyses tools can detect and quantify all major known pre-rRNA species. In addition, they identify several previously-unknown pre-rRNA processing sites. In the final section of the manuscript, they report the mapping of pseudouridylated sites in different pre-RNA species, and conclude that such sites are modified co-transcriptionally.

Given the limitations of current methods for analyzing ribosome maturation, the approaches described here might have high potential for broader application and more in-depth characterization of the pathway. However, there are a few points that, in my opinion, need to be addressed.

My concerns are mostly related to the demonstrations that NanoRibolyzer has sufficient accuracy and sensitivity to quantify various pre-rRNA species. This is essential for detecting altered patterns of ribosome maturation, such as those observed in ribosomopathies.

1. Quantifications of pre-rRNA species and patterns of pre-rRNA contents.

The pre-rRNA species most efficiently detected by nanopore cDNA sequencing (Figures 2B and S4B) are 21S, 21S-C, 12S, and 7S. However, these are not the most abundant species in HEK293 cells. Based on published Northern blot data (and also based on the control samples in Figure 4B), the most abundant pre-rRNAs are 45S/47S, 30S, and 32S. NanoRibolyzer detects well the shortest species but has a reduced sensitivity for longer precursors. In my view, this limitation should be explicitly mentioned in the results section, pointing that the method does not accurately reflect the relative abundance of all pre-rRNA species. This limitation does not preclude the use of the method for the purposes shown in the study, but the extended information provided by Northern blots must be acknowledged.

Northern blots of total RNA (5'-ETS, ITS1 and ITS2 probes) from total cell, nucleus and cytoplasm should be shown for comparison with the results of pre-rRNA contents obtained with NanoRibolyzer.

2. Underestimation of 18S-E contents.

In Figures 2B and S4B, nuclear 18S-E contents appear lower than those of 21S and 21S-C; and cytoplasmic 18S-E is barely detectable. However, it is already known that total 18S-E levels are comparable to or even slightly higher than 21S levels (seen in previously published Northern blot analyses of total RNA from HEK293 cells and in nuclear-cytoplasmic pre-rRNA fractionation data from the Gleizes lab). This discrepancy is also apparent in the control samples of Northern blots in Figure 4B. The reduced detection of 18S-E, particularly in the cytoplasmic fraction, is found with the two fractionation methods used in the study. In my opinion, this problem might be due to the progressive 3'-end trimming of 18S-E during nuclear export described by Preti et al. (2013). To address this, a re-analysis of cytoplasmic RNA-seq data that quantifies, as a pool, the 3'-end-shortened forms of 18S-E could improve recovery and more accurately reflect its true abundance. Accurate quantification of this pool is critical for assessing defects in cytoplasmic maturation of the small ribosomal subunit. Aside from improving its detection in unperturbed cells, this point would be further strengthened by including a knockdown of a protein known to cause cytoplasmic accumulation of 18S-E (for example RPS26) in Figure 4. Demonstrating that

NanoRibolyzer reliably detects this phenotype would validate its utility for studying 40S cytoplasmic maturation defects.

3. Monitoring pre-rRNA processing perturbations.

A complete set of Northern blot analyses, with the different probes for pre-18S and pre-28S precursors, should be provided to demonstrate that all changes detected by NanoRibolyzer are also observed by Northern blot (they might go as supplementary data).

4. Refinement of processing sites and discovery of new pre-rRNA species.

It would be very informative to include data on the relative prevalence of each species.

For instance, among the 30S variants (30S-E2, 30S-C, 30SS, and 30SL), which are the most abundant?

A similar analysis could be useful for the 21S variants (21SS and 21SL) in order to be taken as a reference of major and minor precursors in future studies.

Minor point. The description of the PSE fractionation method used in the final section of the study may not be entirely accurate. Based on the original publication, my understanding is that the SN3 fraction contains preribosomes undergoing early maturation in the nucleolus, the SN2 fraction includes preribosomes at intermediate nucleolar maturation stages, and the SN1 fraction contains the preribosomes in the nucleoplasm and cytoplasm.

(Remarks on code availability)

Reviewer #2

(Remarks to the Author)

In this study, Pastore et al., harness Nanopore sequencing to study rRNA processing and modifications in human cell culture. In principle it is an interesting idea to 1) re-explore rRNA maturation and modifications with this technology and 2) provide a pipeline enabling to study rRNA maturation and modifications. However, the manuscript is rather technical and limited in its biological scope (there is not much new biology discovered). Another serious limitation of this work is that it also lacks conceptual novelty and scientific rigour (among other several novelty claims that are not grounded).

Specific comments:

1) Novelty, references to key previous studies and methodological limitations:

The authors were doing a poor job of properly acknowledging key previous studies.

Among others, studies from the Novoa laboratory, like <https://doi.org/10.1038/s41587-021-00915-6> examining pseudouridylation in (r)RNA are ignored.

Study from the Ferreira-Cerca and Grohmann groups are also ignored <https://doi.org/10.1261/rna.079636.123>. This study has previously demonstrated the use of Nanopore sequencing for the de novo determination of rRNA maturation pathways. Moreover, this study also provided proof of concept to determine precursor-dependent analysis of base modifications. Two previously published key concepts that are claimed by the authors as novelties in this manuscript.

There are also no efforts to critically evaluate the limitations of RiboAnalyzer or how it compares/adds up to what is already existing in the field.

2) rRNA processing site mapping and maturation analysis:

It is not clear what is new or not new regarding the rRNA processing sites. Presentation efforts should be done here.

Previously known rRNA maturation sites and known positions should be better summarized (e.g previous studies vs this study). Please provide the nucleotide positioning when previously determined in comparison to the current study...

The importance of using cellular fractionation to analyse rRNA maturation defects is not clear or justified, especially in the context of analysing ribosome biogenesis factors with known nuclear phenotypes. My impression is that the whole cell analysis does not allow to faithfully capture rRNA maturation in the experimental conditions described (insufficient numbers of reads?). This seems to be expressed indirectly by the authors: "We often observed that whole cell fractions were largely redundant with cytoplasmic fractions". Meaning: no nuclear precursors are observed when doing total RNA sequencing?! If so, it is a major bottle neck. In this regard the Cyt 18S-E recovery seems to me rather poor/modest (and sometime not used or ignored without justification, see point 3). It would be also crucial to look at conditions where this precursor is accumulating in the cytoplasm to demonstrate that this type of events can be faithfully capture.

The author should provide the relevant results and provide clarity on optimal conditions to analyse rRNA maturation (whole cell vs fractionation ...). Please provide experimental evidence and comments to better guide the potential users of the method.

3) Precursor-dependent analysis of base modifications:

The Lafontaine laboratory has previously reported that the m7G and dimethylations modifications are added in the nucleus in human cells <https://doi.org/10.1091/mbc.E15-02-0073>. Accordingly, one would expect to detect these base modifications at the level of Nuc 18S-E and particularly at the level of the downstream Cyt 18S-E, which does not seem to be the case. It is also very unfortunate that the Cyt 18S-E precursor, on which the acp3 modification is also expected to be added, is not used/shown in this analysis, please add.

Overall, the analysis does not provide evidence for the following claim, "In conclusion, NanoRibolyzer pipeline can assign base modification status to particular pre-rRNA precursor based on raw signal analysis.". It is currently an all or nothing comparison without seeing anything clear on the relevant expected pre-rRNA, for instance nuc/cyt18S-E, vs other pre-rRNA intermediates in comparison to "final modification" state (mature 18S). As such these results are useless.

4) Ribosomal RNA pseudouridine formation is mostly cotranscriptional

The conclusion of this analysis is strongly misleading, since the primary 47S rRNA intermediates is NOT a co-transcriptional intermediate. Considering the primary transcript as equal to a co-transcriptional event is an over-simplification and an over-interpretation on the true nature of this pre-rRNA intermediate (where is the evidence of its association with actively transcribing RNA Pol I !!??). At best, the authors can only conclude that most of the pseudouridylation are added rather early.

Some pseudouridylations seem to have a rather "erratic behavior" between precursors. Some residues appear to have lower "signals" than the upstream precursor (e.g. positions 36, 649, 681, 1244, 16943, 1692), instead of being constant or increasing. Please comment.

The density map visualization is also misleading, the darker being the lower modification ratio percentage, the opposite would be more intuitive.

There is also no information for Nuc-18S-E?! Please provide.

A similar depiction related to 28S rRNA pseudouridylation should also be provided somewhere.

The relative percentage of Pseudouridylation varies from site to site, and not all are attaining ~100% within the mature rRNA. I believe that there is information related to their true relative abundance. Please comment on those numbers and probable limitations and relevance. I believe a difference in % between mature rRNA and pre-rRNA (like 47S) would probably provide a clearer view on "late" modifications that may increase over time (if any existing). (The true early modifications should remain constant across the precursors).

(Remarks on code availability)

I could not access the code on Github.

Reviewer #3

(Remarks to the Author)

The 18S, 28S, and 5.8S rRNA species are derived from a common 47S precursor RNA, which is synthesized by RNA polymerase I in the nucleolus. The maturation of these rRNA species from the precursor involves a series of step-wise cleavage and processing events that occur in the nucleolus, nucleoplasm, and cytoplasm. This process generates distinct rRNA intermediates and requires the involvement of numerous trans-acting factors. Although rRNA maturation has been extensively studied over the past 15-20 years, the methodology for defining rRNA intermediates and specific cleavage sites during the maturation process has not seen significant advancements and still primarily relies on Northern blotting and pulse-chase labeling.

Here the authors have developed NanoRibolyzer, an innovative and straightforward rRNA sequencing approach based on long-read nanopore sequencing. They have meticulously benchmarked this method and demonstrated its ability to detect known and unknown pre-rRNA species and pinpoint cleavage sites with single-nucleotide precision. Furthermore, the authors have shown that NanoRibolyzer can detect distinct rRNA modifications as well as alterations in rRNA processing upon depletion of specific processing factors. Overall, this methodology represents a significant advancement for researchers in the field of ribosome biogenesis.

The data in the manuscript are well-organized and clearly presented, and the NanoRibolyzer technology is described in sufficient detail. My only concern is that due to its very technical focus the data remain at a merely descriptive level. For example, the functional significance of the newly identified cleavage sites and rRNA intermediates is not clear. Regardless of this potential weakness, I believe that the technology described here will have a significant impact on the field.

(Remarks on code availability)

As far as I see the code was not connected to material that is related to this manuscript.

Version 1:

Reviewer comments:

Reviewer #1

(Remarks to the Author)

The authors have adequately addressed all the concerns raised during the first round of review. I recommend the manuscript for publication in Nature Communications.

(Remarks on code availability)

Reviewer #2

(Remarks to the Author)

The manuscript has greatly improved but I will reiterate some of my criticisms to the authors. While the specific "NanoRibolyzer" pipeline is somewhat new, the underlying technology of using long-read nanopore sequencing to investigate rRNA modifications and rRNA processing is not entirely novel. Pioneering studies in archaea and other organisms have already demonstrated the utility (and challenges) of long read sequencing for simultaneously identifying the 5' and 3' ends of (r)RNAs and mapping modifications. This study builds upon these established capabilities rather than introducing the concept from scratch. Even though I am convinced that the authors have done a great job developing and benchmarking "Nanoribolyzer" to study human ribosome biogenesis, the manuscript has a certain propensity to over-claim novelty and to not fully acknowledge key previous studies, something I feel unnecessary and that could be avoided without modifying the true impact of this work. Even if not intentional, this reviewer aims to express this concern to the authors. I also have some additional comments:

- 1) Analysis of 18S-E has now improved and since the Gleizes laboratory has nicely demonstrated the addition of non-templated nucleotides during the maturation of 18S-E, the authors should indicate whether these events can be captured by Riboanalyzer? Please provide information in the manuscript.
- 2) Intro: I am not sure what is the intention of the authors regarding the following sentence and the associated references, "Nanopore sequencing (nanopore-seq) has emerged as a promising technology to investigate ribosome biogenesis 15,16". Unless I have overseen this, there is nothing related to ribosome biogenesis in Ref. 15-16, so I am not sure why these are used in this context. Or is it supposed to be a deduction from these references? I believe there are pioneering references in the literature demonstrating proof-of-concept that would be a better fit here.
- 3) Ref. 57 is about rRNA modification analysis and Ref. 58 is about rRNA processing and stage-dependent rRNA modification analysis, however, in its current form, these two references are only associated with rRNA processing analysed by nanopore sequencing. Please correct.
- 4) Last sentence in the discussion: "... with broad applications in basic research and in the study of disease mechanisms, including ribosomopathies and tumorigenesis, as well as in clinical diagnostics." Considering the cellular fractionation step and the RNA amounts currently required, these aspects particularly limit applications in clinical diagnostics.

(Remarks on code availability)

Point-by-point responses to the reviewers' comments:

Reviewer #1 (Remarks to the Author):

This study presents a novel technical approach for quantification and characterization of pre-rRNA intermediates using long-read nanopore sequencing. The authors provide a comprehensive set of validation data showing that their RNA fractionation, sequencing and analyses tools can detect and quantify all major known pre-rRNA species. In addition, they identify several previously-unknown pre-rRNA processing sites. In the final section of the manuscript, they report the mapping of pseudouridylated sites in different pre-RNA species, and conclude that such sites are modified co-transcriptionally.

Given the limitations of current methods for analyzing ribosome maturation, the approaches described here might have high potential for broader application and more in-depth characterization of the pathway. However, there are a few points that, in my opinion, need to be addressed.

Answer: We are grateful to Reviewer #1 for their overall extremely positive assessment of our work.

My concerns are mostly related to the demonstrations that NanoRibolyzer has sufficient accuracy and sensitivity to quantify various pre-rRNA species. This is essential for detecting altered patterns of ribosome maturation, such as those observed in ribosomopathies.

Answer: We thank Reviewer #1 for this comment.

In terms of **accuracy**, our method achieves single-nucleotide resolution, which is the highest possible.

In terms of **sensitivity**, NanoRibolyzer is currently effective with as little as 1 µg of RNA, which is approximately five times less than what is typically required for Northern blotting. In future work, we aim to further reduce the input requirements. However, we hope Reviewer #1 will agree that such further optimization is beyond the scope of the present study.

1. Quantifications of pre-rRNA species and patterns of pre-rRNA contents.

The pre-rRNA species most efficiently detected by nanopore cDNA sequencing (Figures 2B and S4B) are 21S, 21S-C, 12S, and 7S. However, these are not the most abundant species in HEK293 cells. Based on published Northern blot data (and also based on the control samples in Figure 4B), the most abundant pre-rRNAs are 45S/47S, 30S, and 32S. NanoRibolyzer detects well the shortest species but has a reduced sensitivity for longer precursors. In my view, this limitation should be explicitly mentioned in the results section, pointing that the method does not accurately reflect the relative abundance of all pre-rRNA

species. This limitation does not preclude the use of the method for the purposes shown in the study, but the extended information provided by Northern blots must be acknowledged.

Answer: This is a very good point, thank you. Agree to some extent. A strict comparison is challenging because Northern blotting and nanopore sequencing differ markedly in sensitivity. Nonetheless, when comparing the signal captured by NanoRibolyzer (e.g., Fig 2B) with that obtained by Northern blotting (new Supplementary Data 1, HEK293 lanes), it is clear that the species mentioned by the referee—21S/21S-C, 12S (7S is not visible here), and also 18S-E—are at least as abundant, if not more abundant, than 47S/45S, 30S, or 32S. We therefore believe our method largely captures the natural distribution of species

We fully agree with Reviewer #1 that NanoRibolyzer, by design, involves selection steps. This is true for all sequencing-based approaches. Such selection arises (i) from the cDNA-based cloning strategy and (ii) from the physical properties of the nanopores themselves. By contrast, Northern blotting is a “blind” method in which RNAs are directly loaded onto a gel without selection. In this sense, we agree that Northern blotting will remain a valuable and appreciated technique, and we have ourselves used it extensively for two decades.

We have added a short paragraph to the Discussion to acknowledge this point. The new paragraph reads as follows:

By design, NanoRibolyzer, like all sequencing-based approaches, entails inherent selection steps arising from cDNA library preparation and nanopore biophysics; by contrast, Northern blotting is a largely “blind” technique in which RNAs are directly resolved without prior selection, and therefore remains a valuable and complementary method.

We would also like to emphasize that the technology is evolving very rapidly. Over the course of this study, improvements in enzymes and the use of DRS steadily enhanced the detection of low-abundance species such as 45S/47S (see Fig 5, upper left panel, where 47S and 45S are readily detected). Detection became so robust that we were even able to characterize two previously unknown species—47S-01 and 47S-02 (see amended text for details).

Northern blots of total RNA (5'-ETS, ITS1 and ITS2 probes) from total cell, nucleus and cytoplasm should be shown for comparison with the results of pre-rRNA contents obtained with NanoRibolyzer.

Answer: Done. Related to previous question. See **Supplementary Dataset 1**. We also included the 3' ETS probe.

2. Underestimation of 18S-E contents.

In Figures 2B and S4B, nuclear 18S-E contents appear lower than those of 21S and 21S-C; and cytoplasmic 18S-E is barely detectable. However, it is already known that total 18S-E levels are comparable to or even slightly higher than 21S levels (seen in previously published Northern blot analyses of total RNA from HEK293 cells and in nuclear–cytoplasmic pre-rRNA fractionation data from the Gleizes lab). This discrepancy is also apparent in the control samples of Northern blots in Figure 4B. The reduced detection of 18S-E, particularly in the cytoplasmic fraction, is found with the two fractionation methods used in the study. In my opinion, this problem might be due to the progressive 3'-end trimming of 18S-E during nuclear export described by Preti et al. (2013). To address this, a re-analysis of cytoplasmic RNA-seq data that quantifies, as a pool, the 3'-end-shortened forms of 18S-E could improve recovery and more accurately reflect its true abundance. Accurate quantification of this pool is critical for assessing defects in cytoplasmic maturation of the small ribosomal subunit.

Answer: Thank you. This is a very interesting point.

From Figs 2B and S4B, it can be seen that the levels of nuclear 21S, 21S-C, and 18S-E are quite similar; in particular, 18S-E and 21S-C levels are nearly identical.

To further resolve the 3'-end maturation steps of 18S, we instructed our analysis pipeline to incorporate sites corresponding to 3'-extended forms of 18S, as described in Preti *et al.* (see new Fig S7).

This re-analysis, suggested by Reviewer #1 (thank you), beautifully resolved 18S-E and its multiple 3'-extended variants as they mature across the nucleus and the cytoplasm.

Aside from improving its detection in unperturbed cells, this point would be further strengthened by including a knockdown of a protein known to cause cytoplasmic accumulation of 18S-E (for example RPS26) in Figure 4. Demonstrating that NanoRibolyzer reliably detects this phenotype would validate its utility for studying 40S cytoplasmic maturation defects.

Answer: We have performed the suggested experiment. RPS26 was depleted, and nuclear and cytoplasmic RNA fractions from knockdown and control cells were sequenced (new Fig S8). As expected, we observed cytoplasmic accumulation of 3'-extended forms of 18S-E (Fig S8F, red signal in the heatmap). We thank the reviewer for this valuable suggestion.

Furthermore, we applied the same deconvolution analysis to perturbed cells (Fig S17D–F), with particularly striking results upon WBSR22 depletion.

3. Monitoring pre-rRNA processing perturbations.

A complete set of Northern blot analyses, with the different probes for pre-18S and pre-28S precursors, should be provided to demonstrate that all changes detected by NanoRibolyzer are also observed by Northern blot (they might go as supplementary data).

Answer: Completely Agree. Thank you. Done. See Supplementary Figures S10-S14.

4. Refinement of processing sites and discovery of new pre-rRNA species.

It would be very informative to include data on the relative prevalence of each species.

For instance, among the 30S variants (30S-E2, 30S-C, 30SS, and 30SL), which are the most abundant?

A similar analysis could be useful for the 21S variants (21SS and 21SL) in order to be taken as a reference of major and minor precursors in future studies.

Answer: Excellent suggestion. Thank you. This is now shown in Fig 17.

-Minor point. The description of the PSE fractionation method used in the final section of the study may not be entirely accurate. Based on the original publication, my understanding is that the SN3 fraction contains preribosomes undergoing early maturation in the nucleolus, the SN2 fraction includes preribosomes at intermediate nucleolar maturation stages, and the SN1 fraction contains the preribosomes in the nucleoplasm and cytoplasm.

Answer: Agree. But, in our hands, the protocol reproducibly enriched the nucleolus, the nucleoplasm, and the cytoplasm, as we thoroughly established with suitable protein marker detection by western blotting. The differences between the excellent original report from the team of Prof. Dosil, which was a great inspiration to us, and our fractionation, may be due to slight differences in cell line used, extract preparation (pestle used etc.).

Reviewer #2 (Remarks to the Author):

In this study, Pastore et al., harness Nanopore sequencing to study rRNA processing and modifications in human cell culture. In principle it is an interesting idea to 1) re-explore rRNA maturation and modifications with this technology and 2) provide a pipeline enabling to study rRNA maturation and modifications. However, the manuscript is rather technical and limited in its biological scope (there is not much new biology discovered). Another serious limitation of this work is that it also lacks conceptual novelty and scientific rigour (among other several novelty claims that are not grounded).

Answer: We respectfully disagree with Reviewer #2's assessment, and note that Reviewers #1 and #3 do not share this view.

We are not entirely clear which aspects of the study are considered to lack scientific rigour. We would welcome more specific guidance from Reviewer #2, as we believe that both the experimental design and the analytical framework are robust, extensively benchmarked, and carefully validated throughout the manuscript.

We also respectfully disagree with the assertion that the work lacks conceptual novelty. NanoRibolyzer introduces a fundamentally new way of interrogating ribosome biogenesis by enabling quantitative, single-molecule-level analysis of pre-rRNA processing intermediates and RNA modifications within individual precursor species. As highlighted by Reviewers #1 and #3, the field has relied for several decades on classical biochemical approaches—such as Northern blotting and pulse-chase labelling—which, while robust and invaluable, are intrinsically low-resolution, require microgram quantities of RNA, and frequently rely on radioactive labelling, thereby limiting scalability and clinical applicability.

Finally, we respectfully disagree with the claim that the biological scope of the study is limited. Using NanoRibolyzer, we uncover multiple layers of new biology, including:

- (i) the assignment of new or refined functions to previously characterised ribosome biogenesis factors (including DIMT1L and URB1);
- (ii) a novel role for the ribosomal protein RPL3 in 3' ETS processing;
- (iii) the identification of previously undescribed 47S pre-rRNA species (e.g., 47S-01 and 47S-02);
- (iv) the confirmation of previously speculated processing sites (such as site 2²) and the discovery of new cleavage sites (including site E² and others);
- (v) direct evidence that pseudouridylation in human cells occurs very early during ribosome biogenesis, already detectable on unprocessed 47S pre-rRNA;
- (vi) the observation that aberrant 34S and 36S-C RNA species are distinctly hypomodified. While it remains to be determined whether hypomodification is a cause or a consequence of aberrant processing, this finding raises the intriguing possibility that RNA modification is actively coupled to RNA surveillance mechanisms.

Together, these findings demonstrate that NanoRibolyzer is not merely a technical advance, but a platform that enables biological insights that were previously inaccessible with existing methodologies.

1) Novelty, references to key previous studies and methodological limitations:

The authors were doing a poor job of properly acknowledging key previous studies.

Among others, studies from the Novoa laboratory, like <https://doi.org/10.1038/s41587-021-00915-6> examining pseudouridylation in (r)RNA are ignored.

Answer: Apologies. Due to space limitation, at initial submission we compacted the text, and could not cite all references. We take this fully on board. We have added these important references.

Study from the Ferreira-Cerca and Grohmann groups are also ignored <https://doi.org/10.1261/rna.079636.123>. This study has previously demonstrated the use of Nanopore sequencing for the de novo determination of rRNA maturation pathways. Moreover, this study also provided proof of concept to determine precursor-dependent analysis of base modifications. Two previously published key concepts that are claimed by the authors as novelties in this manuscript.

Answer: All apologies for this oversight. These important reference were added. Thank you.

There are also no efforts to critically evaluate the limitations of RiboAnalyzer or how it compares/adds up to what is already existing in the field.

Answer: Thank you. See response above to reviewer #1 and new Discussion.

2) rRNA processing site mapping and maturation analysis:

It is not clear what is new or not new regarding the rRNA processing sites. Presentation efforts should be done here. Previously known rRNA maturation sites and known positions should be better summarized (e.g previous studies vs this study). Please provide the nucleotide positioning when previously determined in comparison to the current study

Answer: We thank Reviewer #2 for their comment. In response, we included Table 1 showing the annotation of pre-rRNA processing sites from three reviews and compared them to our reported sites (all in one referenced pre-rRNA template). More detailed information is presented in Figure S6. It is important to note that rRNA annotations can vary positionally due to sequence heterogeneity or incomplete rRNA annotation. Therefore, it is crucial to accurately document the processing sites with nucleotide resolution using a standard reference sequence.

The importance of using cellular fractionation to analyse rRNA maturation defects is not clear or justified, especially in the context of analysing ribosome biogenesis factors with known nuclear phenotypes. My impression is that the whole cell analysis does not allow to faithfully capture rRNA maturation in the experimental conditions described (insufficient numbers of reads?).

Answer: Indeed, this is correct, and this is because of the large amounts of mature cytoplasmic RNAs which are several days stable and preclude detection of short-lived nuclear precursors.

This seems to be expressed indirectly by the authors: “We often observed that whole cell fractions were largely redundant with cytoplasmic fractions”. Meaning: no nuclear precursors are observed when doing total RNA sequencing?! If so, it is a major bottle neck. In this regard the Cyt 18S-E recovery seems to me rather poor/modest (and sometime not used or ignored without justification, see point 3). It would be also crucial to look at conditions where this precursor is accumulating in the cytoplasm to demonstrate that this type of events can be faithfully captured.

Answer: Thank you. Following the suggestion of Reviewer #1, we have quantified exonucleolytic products, previously hard to measure. Using this approach, we showed the difference in the 18S-E intermediates with a new perturbed factor RPS26 which showed accumulation in cytoplasmic 18S-E variants but not in the nucleus. Similarly, we increased the resolution of the 18S-E intermediates accumulation in WBSR22 depletion, increasing further the sensitivity of the tool to quantify potentially metastable pre-rRNA fragments.

The author should provide the relevant results and provide clarity on optimal conditions to analyse rRNA maturation (whole cell vs fractionation ...). Please provide experimental evidence and comments to better guide the potential users of the method.

Answer: We have provided a complete set of northern blots for all of the tested condition in this study (with all the relevant probes) in supplementary Data 1 and Figs S8-S14 depletions.

3) Precursor-dependent analysis of base modifications:

The Lafontaine laboratory has previously reported that the m7G and dimethylations modifications are added in the nucleus in human cells <https://doi.org/10.1091/mbc.E15-02-0073>. Accordingly, one would expect to detect these base modifications at the level of Nuc 18S-E and particularly at the level of the downstream Cyt 18S-E, which does not seem to be the case. It is also very unfortunate that the Cyt 18S-E precursor, on which the acp3 modification is also expected to be added, is not used/shown in this analysis, please add.

Answer: With respect to m¹acp³Ψ, we first demonstrate that position 1248 on the 18S rRNA is detected in whole-cell and cytoplasmic fractions but, strikingly, not in nuclear fractions

(Fig. 1C, arrow). This observation is fully consistent with the fact that $m^1acp^3\psi$ formation is completed in the cytoplasm.

Regarding the findings of Zorbas et al. (2015), the modification at this site was originally predicted based on differential primer extension; however, this assay did not include nucleo-cytoplasmic fractionation, and therefore could not address the subcellular specificity of modification.

Finally, after categorizing the exonucleolytic processing products of 18S-E as suggested by Reviewer #1, we performed precursor-specific raw signal analysis and identified interesting alterations among the different precursors (see figures below).

For instance, in the case of m^7G , the nuclear 18S-E+78 precursor shows a signal similar to the unmodified 18S IVT, suggesting that it is not modified, whereas the other precursors display distinct patterns.

For $m^1acp^3\psi$, we again observed clear separation between the different precursors and the mature 18S rRNA (relative to the IVT control). We suspect that, because $m^1acp^3\psi$ consists of three individual chemical modifications, these intermediate states likely reflect stepwise maturation, illustrated by the difference between the 18S IVT (yellow), the mature cytoplasmic 18S (red), and the remaining precursors.

As for Nuc-E+9 and Cyt 18S-E+9, we believe that we have reached the limit of our nucleo-cytoplasmic purification protocol and we don't want to over interpret our data. We hope the referee will agree with us, it's a wise decision.

Overall, the analysis does not provide evidence for the following claim, “In conclusion, NanoRibolyzer pipeline can assign base modification status to particular pre-rRNA precursor based on raw signal analysis.”. It is currently an all or nothing comparison without seeing anything clear on the relevant expected pre-rRNA, for instance nuc/cyt18S-E, vs other pre-rRNA intermediates in comparison to “final modification” state (mature 18S). As such these results are useless.

Answer: We are unsure we understand where the reviewer is heading here. We do provide individual species RNA mapping, and this based on raw signal (Fig S19). We do, on the other hand, change the sentence to “In conclusion, NanoRibolyzer pipeline can assign **nucleotide** modification status to particular pre-rRNA precursor based on raw signal analysis.” (Page 8, Line 386)

4) Ribosomal RNA pseudouridine formation is mostly cotranscriptional

The conclusion of this analysis is strongly misleading, since the primary 47S rRNA intermediates is NOT a co-transcriptional intermediate. Considering the primary transcript as equal to a co-transcriptional event is an over-simplification and an over-interpretation on the true nature of this pre-rRNA intermediate (where is the evidence of its association with actively transcribing RNA Pol I !!??). At best, the authors can only conclude that most of the pseudouridylation are added rather early.

Answer: Thank you for this important clarification. We agree with the reviewer that equating the primary 47S transcript with a co-transcriptional intermediate was an incorrect use of terminology, and we regret this imprecision.

We have revised the Results section accordingly, now stating: “A key conclusion from the precursor-specific modification mapping is that Ψ sites are already largely present on the primary 47S transcript” (page 9, line 456).

We have also revised the corresponding text in the Discussion to remove any implication of co-transcriptional modification, now reading: “...we found that many of the modifications present on mature rRNAs were already detectable on the primary transcript (47S), indicating early modification events” (page 10, line 542).

These changes ensure that our conclusions are limited to the timing of modification relative to pre-rRNA processing, without implying direct association with actively transcribing RNA polymerase I.

Some pseudouridylation seem to have a rather “erratic behavior” between precursors.

Some residues appear to have lower “signals” than the upstream precursor (e.g. positions 36, 649, 681, 1244, 16943, 1692), instead of being constant or increasing. Please comment.

Answer: As noted in Supplementary Note 2, our data reliably identified all previously reported Ψ sites (see Taoka et al. (2018)). However, the precise quantification of Ψ levels remains challenging, at this stage. Several studies have shown that Ψ residues can induce characteristic U-to-C mismatches during sequencing (Begik et al., 2021; Tavakoli et al., 2023; Makhamreh et al., 2024; Hewel et al., 2024; Schartel et al., 2024), which can influence both detection sensitivity and the accuracy of modification calling. Additionally, recent work has demonstrated that accurate quantification of pseudouridylation by ONT direct-RNA sequencing strongly depends on the sequence context around the modification and that signal deviations from Ψ extend beyond the canonical 5-mer window (Makhamreh et al., 2024). When combined with our precursor-specific modification analysis, these effects could result in apparent alterations in modification patterns that, at this stage, cannot conclusively interpret as technical artefacts or genuine biological differences.

Despite these limitations, two important observations emerge from our analysis.

- 1) First, we were able to annotate all known pseudouridine sites with high confidence, showing that the primary 47S transcript is reproducibly modified at an early stage.
- 2) Second, the aberrant 34S and 36S-C intermediates are markedly hypomodified, raising interesting hypothesis for future exploration.

We anticipate that ongoing improvements in basecalling models and modification-aware algorithms will progressively overcome current limitations, enabling more accurate and readily accessible datasets for studying RNA modification dynamics.

The density map visualization is also misleading, the darker being the lower modification ratio percentage, the opposite would be more intuitive.

Answer: We thank the reviewer for this suggestion. However, we prefer to keep the heat map in its current format, as the comment concerns mainly a visual/aesthetic aspect. To facilitate interpretation, we have included a detailed figure legend adjacent to the figure to guide the reader. We would also like to emphasize that we have used the same color scheme consistently throughout the manuscript, where: Blue/black represents precursor quantification (reads per million, \log_{10}), Red/black represents pseudouridine modification ratios (in %), and Blue/red indicates the comparison between control and knockdown conditions (\log_2 fold change).

This consistency ensures that readers can easily interpret and compare the data across all figures.

There is also no information for Nuc-18S-E?! Please provide.

Answer: Please find the requested figure below, which has been included in Fig 5D.

A similar depiction related to 28S rRNA pseudouridylation should also be provided somewhere.

Answer: We thank Reviewer 2 for their suggestion. We have now performed a similar analysis focusing on precursor-specific RNA modifications involved in 28S rRNA biogenesis, and we have included in vitro-transcribed 28S RNA as a negative control. Remarkably, we

identified that the aberrant 36S-C species, similar to 34S in 18S biogenesis, was also hypomodified. These observations raise interesting question regarding the role of precursor modification and potential surveillance roles

The relative percentage of Pseudouridylation varies from site to site, and not all are attaining ~100% within the mature rRNA. I believe that there is information related to their true relative abundance. Please comment on those numbers and probable limitations and relevance.

Answer: We have already included in Source Data 3 the detailed modification ratios for each individual site and replicate (demonstrating high reproducibility across replicates) as well as the reference modification ratios reported by Taoka et al., 2018 for comparison (see example below).

Modification ratio (Taoka et al. 2018)	pseudoU Shared/Not Shared with Taoka et al. 2018	Other U-related modifications	Cyt		
			R1	R2	R3
100	Shared		0.730519	0.732012	0.735076
82	Shared		0.676179	0.673204	0.668998
87	Shared		0.857899	0.890407	0.890995
99	Shared		0.594934	0.57908	0.579908
99	Shared		0.492113	0.533851	0.534188
94	Shared		0.307691	0.249198	0.252373

Regarding the accurate quantification and their limitations, we would like to highlight three main points:

- 1) As stated above and in Supplementary note 2, DRS and the dorado basecaller correctly identified pseudouridine sites, but the quantitative ratios should be interpreted with caution, as nearby modifications and local sequence context may affect base calling. We believe that future improvements in DRS base calling accuracy and stoichiometry estimation will improve these issues.
- 2) Although Taoka et al., 2018 serves as the “gold standard” for mapping human rRNA modifications, subsequent studies have reported variation in modification ratios across specific sites and between different cell types (Barozzi et al. 2022, Milenkovic et al. 2025).
- 3) In fact, Milenkovic et al. 2025 recently showed that ribosomal RNA modification levels can vary across different cell types, indicating that direct comparison of modification ratios between Taoka et al. (2018; TK6 and HeLa) and our data (HEK293) is not unexpected.

I believe a difference in % between mature rRNA and pre-rRNA (like 47S) would probably provide a clearer view on “late” modifications that may increase over time (if any existing). (The true early modifications should remain constant across the precursors).

Answer: We thank the reviewer for their suggestion. As requested, we calculated the difference in modification ratios between each precursor and the mature 18S rRNA (per-site) and the results are shown below. However, we believe that presenting it in detail at this stage is not appropriate for the current study. As mentioned above, although we can reliably identify pseudouridine sites, their biological significance remains difficult to interpret and lies beyond the scope of this work.

Reviewer #2 (Remarks on code availability):

I could not access the code on Github.

Answer: We apologize for this. We have created a token that allows the reviewer to access the repository in a read only format. To download the code please follow these instructions in the command line.

```
git clone https://github.com/stegiopast/wf-nanoribolyzer.git
```

Username: stegiopast

Password:

```
github_pat_11ANNHK4Q0ACiq2v0l7uiF_C2FEZwDT8b5jSZzIOEw6YSqIHSBfs6puMIH  
X9H0S096QRXFUFD60dHDD5qi
```

Once the repository is downloaded one can download epi2me and place the repository folder into /home/user/epi2me/workflows/foo_folder/

When placed in the right folder the pipeline should work as long as the machine has a GPU access and the nvidia-container toolkit, nextflow and docker are installed. These are standard installations to set up Epi2Me.

Reviewer #3 (Remarks to the Author):

The 18S, 28S, and 5.8S rRNA species are derived from a common 47S precursor RNA, which is synthesized by RNA polymerase I in the nucleolus. The maturation of these rRNA species from the precursor involves a series of step-wise cleavage and processing events that occur in the nucleolus, nucleoplasm, and cytoplasm. This process generates distinct rRNA intermediates and requires the involvement of numerous trans-acting factors. Although rRNA maturation has been extensively studied over the past 15-20 years, the methodology for defining rRNA intermediates and specific cleavage sites during the maturation process has not seen significant advancements and still primarily relies on Northern blotting and pulse-chase labeling.

Here the authors have developed NanoRibolyzer, an innovative and straightforward rRNA sequencing approach based on long-read nanopore sequencing. They have meticulously benchmarked this method and demonstrated its ability to detect known and unknown pre-rRNA species and pinpoint cleavage sites with single-nucleotide precision. Furthermore, the authors have shown that NanoRibolyzer can detect distinct rRNA modifications as well as alterations in rRNA processing upon depletion of specific processing factors. Overall, this methodology represents a significant advancement for researchers in the field of ribosome biogenesis.

Answer: We are grateful for a positive assessment of our work.

The data in the manuscript are well-organized and clearly presented, and the NanoRibolyzer technology is described in sufficient detail.

Answer: Again, we thank Reviewer #3 for their positive evaluation.

My only concern is that due to its very technical focus the data remain at a merely descriptive level. For example, the functional significance of the newly identified cleavage sites and rRNA intermediates is not clear. Regardless of this potential weakness, I believe that the technology described here will have a significant impact on the field.

Authors: Firstly, we truly appreciate Reviewer #3's positive comments and their recognition of the potential impact that NanoRibolyzer can have in the field in years to come.

We somehow disagree, since in addition to establishing a completely novel pipeline of analysis, we also reveal:

- new functions for known factors (case of DIMT1L for example, known to be important for ITS1 processing, for which we show it is also involved in earlier and later maturation steps)
 - new functions for ribosomal proteins (3' ETS processing for RPL3)
 - observation of unique degradation rRNA fingerprints, with biological values since observed in cells depleted of factors involved in the same maturation steps (URB1 and RPL3)
- All these elements are novel biology, and not purely technique.

Reviewer #3 (Remarks on code availability):

As far as I see the code was not connected to material that is related to this manuscript.

Answer: Similar to reviewer 2, we provide here the token to access the code. To download the code please follow these instructions in the command line.

```
git clone https://github.com/stegiopast/wf-nanoribolyzer.git
```

```
Username: stegiopast
```

```
Password:
```

```
github_pat_11ANNHK4Q0ACiq2v0l7uiF_C2FEZwDT8b5jSZzIOEw6YSqIHSBfs6puMIH  
X9H0S096QRXFUFD60dHDD5qi
```

Point-by-point responses to the reviewers' comments:

Reviewer #1 (Remarks to the Author):

The authors have adequately addressed all the concerns raised during the first round of review.

I recommend the manuscript for publication in Nature Communications. (Remarks on code availability)

Answer: We sincerely thank Reviewer #1 for their valuable suggestions, which have helped us improve the quality of our work. Thank you!

Reviewer #2 (Remarks to the Author)

The manuscript has greatly improved but I will reiterate some of my criticisms to the authors. While the specific "NanoRibolyzer" pipeline is somewhat new, the underlying technology of using long-read nanopore sequencing to investigate rRNA modifications and rRNA processing is not entirely novel. Pioneering studies in archaea and other organisms have already demonstrated the utility (and challenges) of long read sequencing for simultaneously identifying the 5' and 3' ends of (r)RNAs and mapping modifications. This study builds upon these established capabilities rather than introducing the concept from scratch. Even though I am convinced that the authors have done a great job developing and benchmarking "Nanoribolyzer" to study human ribosome biogenesis, the manuscript has a certain propensity to over-claim novelty and to not fully acknowledge key previous studies, something I feel unnecessary and that could be avoided without modifying the true impact of this work. Even if not intentional, this reviewer aims to express this concern to the authors.

Answer: Thank you. We have already introduced in our manuscript the references to which Reviewer #2 is alluding. We hope Reviewer #2 understands that this is a research manuscript, and therefore there is limited space to compare in detail every method previously developed.

We are nevertheless grateful to Reviewer #2, as these comments have prompted us to plan a review article in which we will compare the available methods in greater detail.

I also have some additional comments:

1) Analysis of 18S-E has now improved and since the Gleizes laboratory has nicely demonstrated the addition of non-templated nucleotides during the maturation of 18S-E, the authors should indicate whether these events can be captured by Riboanalyzer? Please provide information in the manuscript.

Answer: We thank Reviewer #2 for their suggestion. While we fully agree that the addition of non-templated nucleotides (such as oligo-adenylation/uridylation), is

important, we believe that a detailed analysis of these mechanisms lies beyond the current scope of our manuscript. As Gleizes's lab demonstrated, mechanisms such as oligo-adenylation and uridylation, play a double role both in processing as well as ribosomal RNA surveillance. While our pipeline does capture such events (data shown below – please keep this unpublished information confidential), we feel that presenting such results is out of scope of the present manuscript. In fact, we recognize the importance of this topic and are actively working to collect further evidence through more targeted experiments. We look forward to addressing this in a dedicated follow-up study.

2) Intro: I am not sure what is the intention of the authors regarding the following sentence and the associated references, “Nanopore sequencing (nanopore-seq) has emerged as a promising technology to investigate ribosome biogenesis 15,16”. Unless I have overseen this, there is nothing related to ribosome biogenesis in Ref. 15-16, so I am not sure why these are used in this context. Or is it supposed to be a deduction from these references? I believe there are pioneering references in the literature demonstrating proof-of-concept that would be a better fit here.

Answer: We thank the reviewer for this comment. The references included in this paragraph (ref 15-18) are focused mostly on introducing Nanopore-seq technology and its unique features of long-read sequencing and RNA modification detection as a general. Therefore, we included two major reviews that we thought summarize nicely this technology. The following sentences describe why is it a promising technology to investigate Ribosome biogenesis “The key advantage of nanopore-seq lies in its ability to sequence long reads, such as cDNAs, as well as native RNA molecules via direct RNA sequencing (DRS), allowing for the investigation of entire transcripts including their 5’ and 3’ ends and their modifications 17,18”. However, we agree with the reviewer that in the case of this sentence, inclusion of a more relevant citation such as Grünberger et al. 2021. Therefore, we have included this citation here.

3) Ref. 57 is about rRNA modification analysis and Ref. 58 is about rRNA processing and stage-dependent rRNA modification analysis, however, in its current form, these

two references are only associated with rRNA processing analysed by nanopore sequencing. Please correct.

Answer: We thank the reviewer for this comment. We have corrected these citations and included both citations for the description of rRNA modification analysis.

4) Last sentence in the discussion: ...” with broad applications in basic research and in the study of disease mechanisms, including ribosomopathies and tumorigenesis, as well as in clinical diagnostics.” Considering the cellular fractionation step and the RNA amounts currently required, these aspects particularly limit applications in clinical diagnostics.

Answer: We thank Reviewer #2 for raising this important point. We respectfully clarify that, although the RNA input required for NanoRibolyzer remains significant, it already represents more than a tenfold reduction compared with Northern blotting, which typically requires microgram quantities of RNA. This reduction enables analysis from hundreds of nanograms of input material, representing a substantial improvement over existing approaches and expanding the range of accessible biological samples.

We agree that the current workflow, including the requirement for cellular fractionation and relatively high RNA input, may presently limit immediate application in routine clinical diagnostics. Nevertheless, NanoRibolyzer offers important practical advantages, including the absence of hazardous radiolabeled reagents and substantially improved resolution, which make it well suited for future clinical implementation.

Importantly, we are actively working to further reduce RNA input requirements through both experimental and computational optimization. Ongoing developments aim to enable robust analysis from low-input samples and may ultimately reduce or eliminate the need for fractionation.